# Night-Time Light Remote Sensing Mapping: Construction and Analysis of Ethnic Minority Development Index

Fei Zhao [1,2], Lu Song [1], Zhiyan Peng [1], Jianqin Yang [3], Guize Luan [1], Chen Chu [1], Jieyu Ding [4], Siwen Feng [5], Yuhang Jing [1] and Zhiqiang Xie [1,2,*]

1   School of Earth Sciences, Yunnan University, Kunming 650500, China; cartographer@ynu.edu.cn (F.Z.); songlu@mail.ynu.edu.cn (L.S.); pengzy834@mail.ynu.edu.cn (Z.P.); luanguize@mail.ynu.edu.cn (G.L.); chuchen2017@mail.ynu.edu.cn (C.C.); yuhang@mail.ynu.edu.cn (Y.J.)
2   Engineering Research Center of Domestic High-Resolution Satellite Remote Sensing Geology for Universities of Yunnan Province, Kunming 650500, China
3   Kunming Municipal People's Congress Standing Committee, Kunming 650500, China; yjqshe@163.com
4   School of Resources and Environmental Science, Wuhan University, Wuhan 430079, China; jadeding@whu.edu.cn
5   Institute of International Rivers and Eco-Security, Yunnan University, Kunming 650500, China; fengsiwen1004@mail.ynu.edu.cn
*   Correspondence: xzq_2019@ynu.edu.cn

**Abstract:** Using toponym data, population data, and night-time light data, we visualized the development index of the Yi, Wa, Zhuang, Naxi, Hani, and Dai ethnic groups on ArcGIS as well as the distribution of 25 ethnic minorities in the study area. First, we extracted the toponym data of 25 ethnic minorities in the study area, combined with night-time light data and the population proportion data of each ethnic group, then we obtained the development index of each ethnic group in the study area. We compared the development indexes of the Yi, Wa, Zhuang, Naxi, Hani, and Dai ethnic groups with higher development indexes. The results show that the Yi nationality's development index was the highest, reaching 28.86 (with two decimal places), and the Dai nationality's development index was the lowest (15.22). The areas with the highest minority development index were concentrated in the core area of the minority development, and the size varied with the minority's distance. According to the distribution of ethnic minorities, we found that the Yi ethnic group was distributed in almost the entire study area, while other ethnic minorities had obvious geographical distribution characteristics, and there were multiple ethnic minorities living together. This research is of great significance to the cultural protection of ethnic minorities, the development of ethnic minorities, and the remote sensing mapping of lights at night.

**Keywords:** night-time light remote sensing; ethnic minorities; core ethnic minority areas; development index; toponym data

## 1. Introduction

Ethnic minorities refer to ethnic groups other than the main ethnic group in a multi-ethnic country. The proportion of their population is smaller than that of the main ethnic group. There are currently more than 2000 ethnic groups in the world, and the total number of Asian ethnic groups is more than 1000, accounting for about half of the total number of ethnic groups in the world. Among them, the total number of ethnic groups in China, India, the Philippines, and Indonesia exceeds 50. There are about 170 ethnic groups in Europe, and there are about 20 basically single-ethnic countries. There are 55 ethnic minorities in China except for the main ethnic group. The distribution of ethnic minorities in China is relatively wide, mainly showing the distribution of "large mixed residences and small settlements". The indicators to measure the development level of a region include education level [1], regional GDP [2–4], population [5,6], poverty index [7], etc. Among them, the most direct and quantifiable one is economic development. The most direct connection between a

nation and a country is the consistency of economic interests [8]. The distribution of ethnic minorities is different, their ecological environment, cultural diversity (such as living habits, languages, religious beliefs, etc.), the technology used in production, the allocation of resources is different, so their economic development is also different [9]. The economic development of ethnic minorities is part of the country's economic development and contributes to the economic development of the entire country. If there is a problem with the economic development of ethnic minorities, it will directly affect the country's economic development to a certain extent. Due to differences in living environment and life concepts, there are different economic development models in economic development, leading to better ethnic development in some places and poorer ethnic development in other places. However, the economic development of China's ethnic minority areas is generally unbalanced. China is a multi-ethnic country, and the common development and mutual assistance of all ethnic groups can make our country stronger and more prosperous. However, due to the different levels of economic development of different ethnic groups, studying the development of ethnic minorities plays an important role in formulating and adjusting corresponding policies. It is very important to understand and discover the development status of each ethnic group. This study helps to understand the development status of ethnic minorities through a simple and quick method.

At present, it mainly studies the economic development index of ethnic minorities from gross domestic product (GDP). A study of the economic development status of the five western ethnic autonomous regions in Inner Mongolia, Guangxi, Tibet, Ningxia, and Xinjiang found that the GDP of the five ethnic minorities regions lagged behind the national level, and there were also significant differences in the economic development level of ethnic minorities in the prefectures regions. The urban–rural per capita income ratio exceeded 2.5:1, and the highest urban–rural per capita income ratio reached 5.6:1, which far exceeded the international standard (according to the general international situation, the per capita GDP is between US$800 and US$1000, and the urban–rural per capita income ratio is 1.7:1 or so) [10]. Li [11] found that the income gap between urban and rural areas in ethnic minorities regions is large, as was the gap between GDP and the national level. The absolute difference in the per capita GDP of the ethnic minorities in Northwest China is gradually expanding, and the absolute difference in the economic development level of the ethnic minorities is expanding [12,13]. Zheng [14] pointed out in his research that both in terms of innovation and economic development, ethnic minority areas lagged behind the national level, and there were large differences in economic development among ethnic minority areas. Luo and Zhuang [15] conducted research on the economic development of the two provinces of Guangxi and Yunnan in the past 15 years, and found that the higher the proportion of the minority population in the total population, the lower the economic development level of the county-level region. Although there are many studies on the development of ethnic minorities, there are very few studies on the development index of ethnic minorities, and the research on the GDP of ethnic minorities only stays at the level of statistical yearbook research and qualitative analysis. The use of more scientific methods to study the development index of ethnic minority regions is of reference significance for understanding the development of ethnic minority regions, the development differences of various ethnic minorities, and the state's formulation of corresponding policies.

Night-time light data refer to the capture of town lights, fishery lights, etc. at night without clouds [16]. The currently widely used night-time light data include: (1) The Defense Meteorological Satellite Program's Operational Linescan System (DMSP-OLS) satellite, which provides data from 1992 to 2013; (2) The Suomi National Polar-Orbiting Partnership's Visible Infrared Imaging Radiometer Suite (NPP-VIIRS), which provides data from 2012 to the present; and (3) China's first professional night-time light remote sensing satellite "Luojia-1", jointly developed and produced by the Wuhan University team and related institutions, which provides data from 2018 to the present.

The level of human activities and economic development can be better reflected by night-time light remote sensing data, so it is widely used in social and economic

fields [17–19] such as economic activity monitoring [20] and economic development research [21]. Doll et al. [22] used night-time light data to assess socio-economic development and found that it was highly correlated with GDP on a national scale ($R^2 = 0.85$, when $R^2$ is greater than 0.8, it can be considered that the two variables are highly correlated), and simulated the spatial distribution of GDP. Elvidge et al. [23] used DMSP-OLS data to analyze the relationship between night lighting area and GDP in 200 countries and found that there was a good linear relationship between night-time light area and GDP. Henderson et al. [24] used a DMSP stabilized light source and radiometric correction images, which correctly reflected the differences in the social and economic development levels of San Francisco, Beijing, and Lhasa. Henderson et al. [25] found that the brightness of night lights in a country had an obvious linear relationship ($R^2 = 0.8$) with the country's GDP development level. Michalopoulos et al. [26] used a similar method (similar to Henderson et al.) to study the correlation between night-time light data and GDP in Africa, and got good results. Wu et al. [27] used DMSP-OLS data to estimate GDP and the results were satisfactory. Jiang et al. [28] used DMSP-OLS data and NPP-VIIRS data to perform regression simulations on multiple socio-economic parameters, and found that using NPP-VIIRS night-time light data to regress with the whole city's GDP, $R^2$ reached 0.9102. This proves that night-time light data have a good linear correlation with GDP and power consumption, and found that NPP-VIIRS had higher accuracy and more advantages. Zhu et al. [29] found that compared with traditional socio-economic indicators (GDP, oil and gas production, etc.), night light data are more sensitive and more intuitively reflects social and economic development.

Some scholars have also used night-time light data to study the poverty index of a region. This method can also reflect the development status of the region to a certain extent. Li et al. [30] used the method of machine learning, combined with the robust features of the night light image spatial characteristics to identify China's high-poor counties. The overall accuracy of the results was greater than 82%, and the user accuracy was greater than 63%. Andreano et al. [31] used DMSP-OLS data to perform spatial classification and continuous time estimation of poverty gap, number of people, and Gini index in 20 Latin American and Caribbean countries. It was found that combining night-time light data helped to better understand poverty and its temporal and spatial dynamics. Pokhriyal et al. [32] used environmental data and call data records to accurately predict the global multidimensional poverty index. This method has high accuracy in predicting health, education, and living standards (Pearson's correlation coefficient is 0.84–0.86). Li et al. [33] used the principal component analysis method to establish a comprehensive multi-dimensional poverty index, and showed the temporal and spatial heterogeneity of multi-dimensional poverty in 2311 counties in China. It was found that the mountainous areas of Southwest, North China, Northwest China, and the plateau areas of Southeast China had higher levels of economic development.

A large number of studies have proven that the night-time light data reflect the development level of a region, so it is feasible to use it to construct a development index. Compared with traditional statistical yearbook research and qualitative analysis, this paper used night-time light data to construct the development index of ethnic minority areas, which is more accurate and saves resources.

## 2. Materials and Methods

### 2.1. Materials

#### 2.1.1. Study Area

Yunnan Province is located on the border of southwestern China. Its geographic location is between 21°8′–29°15′ N and 97°31′–106°11′ E. Yunnan Province is the province with the largest number of ethnic minorities in China. According to the statistics of the sixth national census in 2010, there are 25 ethnic minorities in Yunnan Province, among which the population of Yi, Bai, and Dai are larger. Among the 25 ethnic minorities in Yunnan Province, 15 ethnic minorities are unique to Yunnan such as the Bai, Hani, Lisu, Dulong, etc.

The development of ethnic minorities in Yunnan Province has made great contributions to the socio-economic development of the entire Yunnan Province. Yunnan Province is a mountainous plateau. Compared with provinces in plain areas, its topographic features are unfavorable for its development. However, at the same time, Yunnan Province is located on the border of southwest China and is a key area for the development of the "Belt and Road" initiative. There are 16 prefecture-level administrative regions in Yunnan Province including eight prefecture-level cities, eight autonomous prefectures, 17 county-level cities, and 129 county-level districts. Among the 16 prefecture-level administrative regions, there are eight ethnic minority core areas. The administrative division and specific geographical location of Yunnan Province are shown in Figure 1.

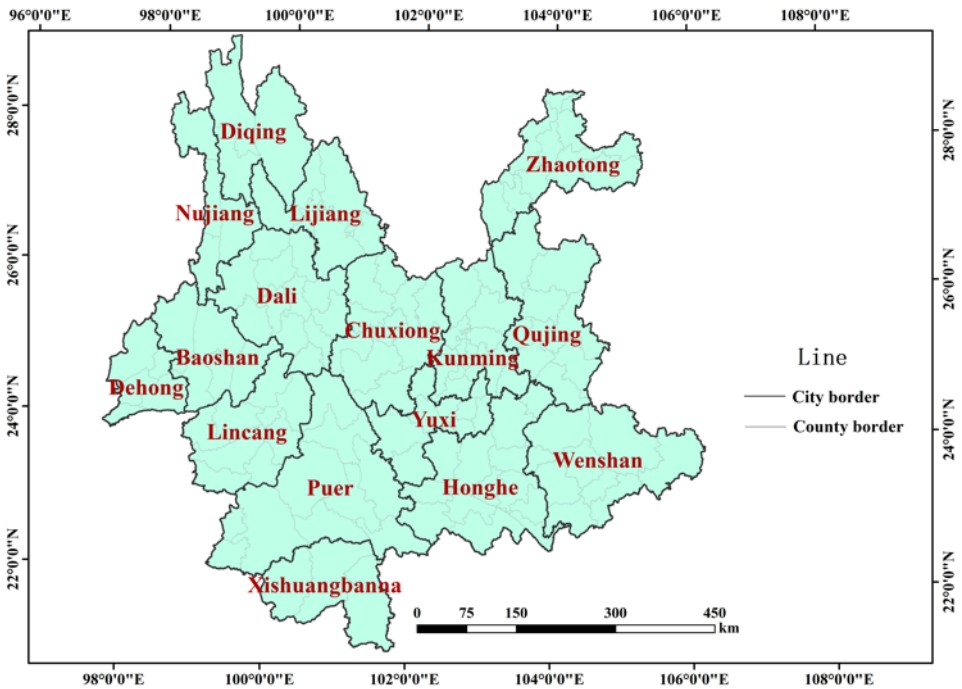

**Figure 1.** Geographical location and administrative boundaries of Yunnan Province.

2.1.2. Data Sources

The data used in this article are as follows (Table 1): (1) NPP-VIIRS composite data; (2) toponym data; (3) Yunnan Province census statistics; (4) Yunnan Province county level Administrative division boundaries; and (5) Yunnan Statistical Yearbook Data.

**Table 1.** Details of the data sources in this study.

| Data | Data Information | Year | Source |
|---|---|---|---|
| NPP-VIIRS | NPP-VIIRS cloudless DNB compound monthly average data | 2018 | Earth Observation Group (EOG) (https://eogdata.mines.edu/download_dnb_composites.html, accessed on 28 May 2020) |
| Toponym | Results of the Second National Toponymic Census of China | 2019 | China National Geographical Names Database (http://dmfw.mca.gov.cn/, accessed on 20 May 2020) |
| Statistics of Yunnan Province Census | Data from the Sixth Census of Yunnan Province | 2010 | China Social Big Data Research Platform (http://data.cnki.net/, accessed on 14 June 2020) |
| Boundaries of county-level administrative divisions in Yunnan Province | County-level vector data in Yunnan Province | 2017 | National Basic Geographic Information Center (http://www.ngcc.cn/ngcc/, accessed on 13 May 2019) |
| Yunnan Statistical Yearbook Data | Socio-economic indicators of Yunnan Province | 2013–2018 | People's Government of Yunnan Province (www.yn.gov.cn, accessed on 2 May 2021) |

The NPP-VIIRS night-time light data adopt the monthly average data of the global cloudless Day–Night Band (DNB) composite data in 2018, and the spatial resolution of NPP-VIIRS data is 500 m. Studies have shown that the DNB of the NPP satellite system is widely used to estimate social and economic parameters, and the in-orbit radiation correction can improve data quality [34,35]. Finally, monthly average data were used to synthesize annual average data for research. The data were downloaded from the Earth Observation Group (EOG) (https://eogdata.mines.edu/download_dnb_composites.html, accessed on 28 May 2020).

The toponym data used in the study come from the results of the second national toponym data census, which mainly includes the meaning of toponyms, that is, the ethnic types of toponyms, the feature type of toponyms, the historical sources of toponyms, the spatial location, and other information, which can be downloaded from the China National Geographical Names Information Database (http://dmfw.mca.gov.cn/, accessed on 20 May 2020).

The census statistics of Yunnan Province use the data of the sixth national census, and the data can be downloaded from the sixth census data of Yunnan Province on the China Social Big Data Research Platform (http://data.cnki.net/, accessed on 14 June 2020). In the data, detailed statistics are made on the population of all ethnic groups in the county-level regions of Yunnan Province.

The county-level administrative divisions of Yunnan Province are derived from the 1:4 million vector data provided by the National Basic Geographic Information Center. In order to make the research more convenient, all the data in this paper were converted into the Lambert projection (Asia_Lambert_Conformal_Conic) based on WGS_1984. In order to make the research more accurate, combined with the geographic location of the study area, the central meridian was set to 102°, the first standard latitude was 22°, and the second standard latitude was 28.3°.

The statistical yearbook data contain a large amount of socio-economic data such as regional GDP per capita, regional total GDP, and regional employees. The development data and production methods of a region can be obtained from the statistical yearbook. The statistical yearbook data of Yunnan Province from 2013 to 2018 was used to verify the feasibility of the method in this paper.

*2.2. Methods*

Using the 2018 NPP-VIIRS night-time light data to construct the Yunnan Minority Development Index requires the following three steps. First, preprocess the downloaded NPP-VIIRS cloudless DNB composite monthly average data to obtain stable night light data. Second, extract the toponym data that contain minority information in the toponym data to obtain the Yunnan Province minority toponym dataset, and conduct a kernel density analysis on each type of ethnic minority toponym data in Yunnan Province. Calculate the minority development index using the results of kernel density analysis combined with the results of the minority population proportion grid results and the NPP-VIIRS night-time light data. Finally, in order to more clearly reflect the distribution of ethnic minorities, combine the toponym data and the results of the minority development index to obtain the research area distribution of 25 ethnic minorities. The specific process is shown in Figure 2.

2.2.1. NPP-VIIRS Data Preprocessing

In order to avoid the influence of grid deformation, sensors, and other factors on the research results, first, geometric correction was performed on the 2018 NPP-VIIRS monthly cloudless DNB composite data using the geometric correction tool in ENVI. Since the geographic coordinate system of the acquired NPP-VIIRS data is WGS_1984, set the projection parameter to the WGS_1984 geographic coordinate system, set the output pixel size to 1000 m, and select the cubic convolution method as the resampling method. The NPP-VIIRS night-time light data obtained include fires, aurora, and other noises. Therefore, it needs to be radiated to eliminate the influence of background noise. The process of

radiant correction can be referred to in [36]. Load the data to be corrected in ENVI and use the RPC orthorectification workflow tool for correction. First, select the average radiance value of the cloud in the low reflectivity area of the sea surface as the calibration value for removing scattered light, and then subtract the calibration value from the entire image to remove the cloud scattering. Second, using the method of adjacent aberrations, a threshold was set to obtain a stable surface area, and the obtained stable surface area was used as a mask, and the radiation value of the mask area was statistically analyzed. Finally, three times the average radiation value of the statistical analysis was taken as the confidence interval to remove the surface scattered light. After radiant correction, effective night-time light data can be obtained. Then, use the data after geometric correction and radiometric correction to synthesize the 2018 annual average data. The calculation formula is:

$$DN_j = \frac{\sum_{i=1}^{12} DN_i}{12},$$ (1)

where $DN_i$ represents the light brightness value in month $i$, and $DN_j$ represents the average light brightness value in year $j$.

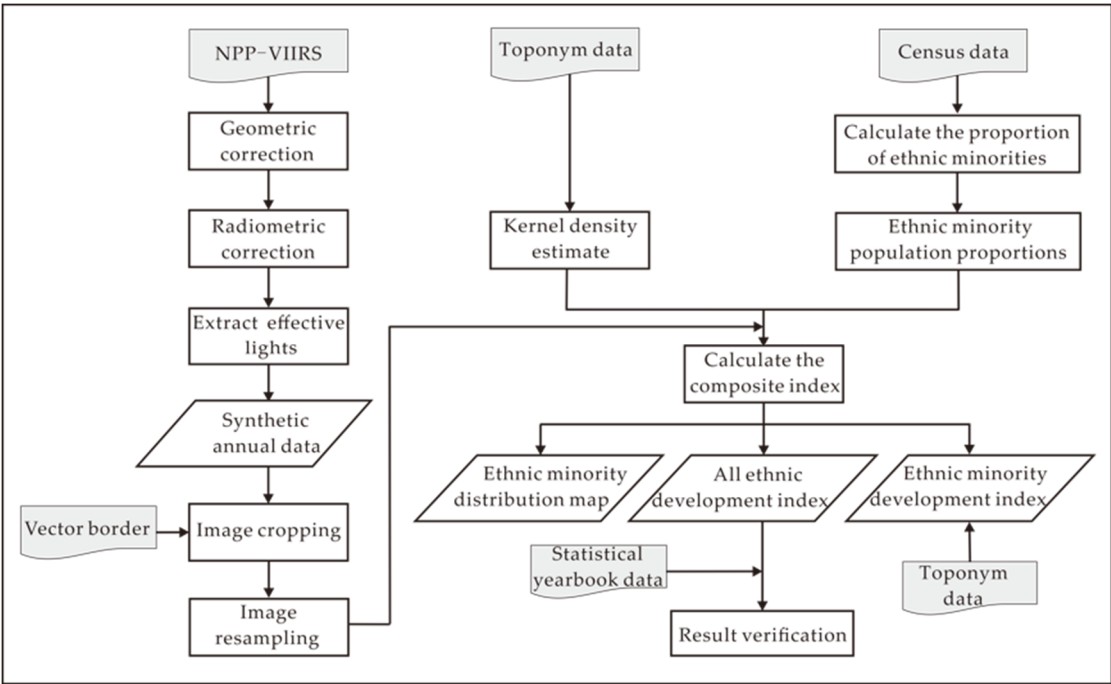

**Figure 2.** Flowchart of the methodology.

After synthesizing the 2018 NPP-VIIRS annual data, we used the administrative divisions of Yunnan Province as a mask to trim the night-time light data to obtain the study area. In order to make subsequent research more convenient, the coordinates were unified into the Lambert projection based on WGS_1984. Finally, using the cubic convolution interpolation method to resample the NPP-VIIRS data to a grid size from the original pixel size of 500 m × 500 m to 1000 m × 1000 m, and obtained stable night light data in 2018. The results are shown in Figure 3.

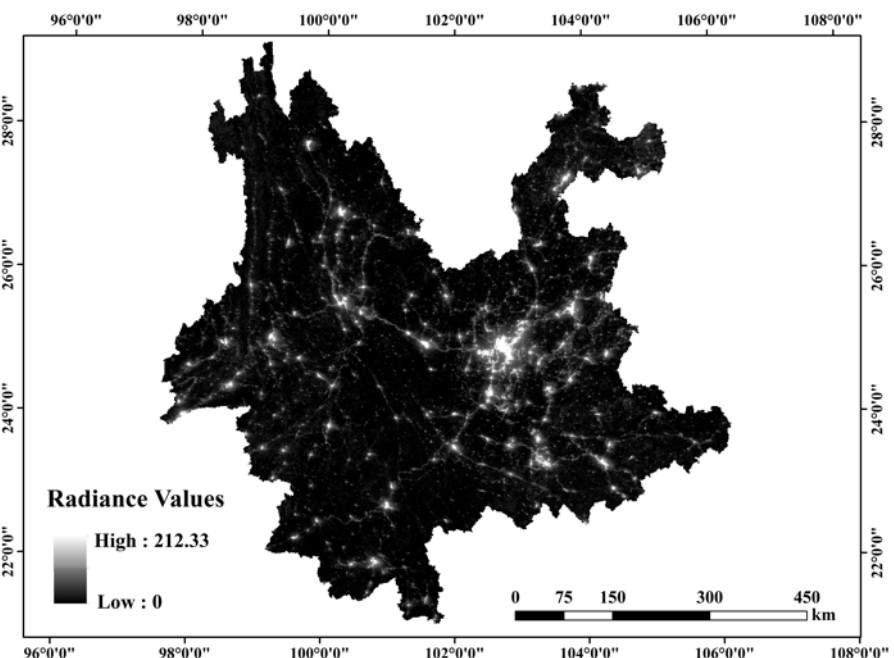

**Figure 3.** Data processing results of NPP-VIIRS in 2018.

### 2.2.2. Construction of the Development Index of Various Ethnic Minorities

　　　Gelling stated that toponyms are "road signs to understand the past" [37] as toponyms are used to indicate the names of specific geographic areas and contain rich information such as the ethnic type of the local residents and the interpretation of the geographical environment by local people at the time of naming [38,39]. Studying toponyms is the basis for understanding the national culture and local characteristics of a region [40]. From toponym data, the ethnic types, language and culture, and religious beliefs of a region [41], spatial location, and the environmental evolution process related to history [42,43], environment, and landforms [44] can be extracted. This is of great significance for understanding ethnic minority settlements and the distribution of ethnic minorities.

　　　The national census is a census about the population of the whole country. The contents of the census mainly include gender, age, ethnicity, etc. The subjects of the census are mainly natural persons living in the People's Republic of China (except Hong Kong, Macau, and Taiwan). From the census data, information about ethnic minorities can be extracted such as the place of residence of the ethnic minority population, and information about the proportion of the ethnic minority population can also be further extracted.

　　　The distribution of ethnic minorities in China mainly shows the distribution of "large mixed residences and small settlements". Therefore, the toponym of ethnic minorities will be unevenly distributed, and the toponym data obtained are discrete measured values. Kernel density estimation (KDE) is used to calculate the unit density of the measured value of point and line elements within a specified area. It can intuitively reflect the distribution of discrete measured values in a continuous area. Kernel density estimation can obtain the weighted average density of all data points in the study area [45]. The weight assigned is related to the distance of the center point of the data point. The farther away from the center point, the smaller the weight is assigned, and vice versa [46]. The formula for calculating the kernel density $P_i$ at any point $i$ in space is:

$$P_i = \frac{1}{n\pi R^2} \times \sum_{j=1}^{n} K_j \left(1 - \frac{D_{ij}^2}{R^2}\right)^2 , \qquad (2)$$

where $R$ is the search radius (bandwidth) of the selected area ($D_{ij} < R$); $K_j$ is the weight of the research data point $j$; $D_{ij}$ is the distance between the space point $i$ and the research data

point $j$; and n is the number of research data points $j$ within the search radius $R$. The search radius $R$ has a direct impact on the results of kernel density analysis [47].

In this study, 25 ethnic minority geographic names were used for kernel density analysis. Because the area of an ethnic minority gathering area in the study area is about one square kilometer. According to this feature, through comparative analysis, the search radius of kernel density estimation is constantly changed, and finally, it was found that when the search radius was 1000 m, the effect was better, and can distinguish ethnic minority gathering areas. Considering that there are places with ethnic minority toponyms, but no ethnic minorities living in them, this paper used census data to calculate the proportion of 25 ethnic minorities in the study area, and obtained a grid map of the proportion of 25 ethnic minorities for future use.

The development of a region or a nation is often affected by many factors such as population, economy, environment, geographical location, etc. In addition, there are differences in the development of different regions of the same ethnic group and between different ethnic groups in the same region. Therefore, it is necessary to construct a development index that can reflect this difference in order to quantitatively analyze the development of ethnic minorities. This article used population, toponym data, and NPP-VIIRS data combined with the literature [48,49] as well as the formula form of the spatialization of population data to propose a method to calculate the development index of various ethnic minorities. The calculation formula is shown in Equation (3):

$$CPS_i = \sqrt{PR_i \times KDE_i \times NPP_i} \tag{3}$$

where $CPS_i$ is the development index of minority $i$; $PR_i$ is the population proportion of minority $i$; $KDE_i$ is the kernel density analysis result of minority $i$; and $NPP_i$ is the night light radiance value of minority $i$.

### 2.2.3. Distribution of Ethnic Minorities

In order to clearly understand the distribution of each ethnic group, we used the obtained ethnic development index combined with ethnic toponym data. We used the 2018 NPP-VIIRS data as a base map, and used the point method to show the distribution of 25 ethnic minorities. Due to the large number of ethnic minorities, it was difficult to distinguish between ethnic groups using only different colors. This paper applied the literature [50] on the classification of language affiliation, and used the language branches of different ethnic minority languages to classify 25 ethnic minorities into 13 categories. Since the 13 categories were difficult to distinguish on the map, the 13 categories were merged into six categories based on the language branch classification. The specific classification is shown in Table 2.

**Table 2.** Language branch classification.

| Branch | Ethnic Minority |
|---|---|
| Yi Branch | Yi, Lisu, Naxi, Bai, Lahu, Hani, Jinuo |
| Zhuang and Dai Branch | Zhuang, Buyi, Dai |
| Tibetan Branch | Tibetan |
| Jingpo Branch | Jingpo, Dulong |
| Chinese Branch | Hui, Manchu |
| Other Languages | Achang (Burmese branch), Shui (Dong Shui branch), Pumi; Nu; Mongolian, Deang (Undecided language), Miao (Miao branch), Yao (Yao branch), Wa; Bulang (Benglong language branch) |

Since the development index of the Yi nationality was the highest, but less than 30, the 0–30 was divided into five categories by the equal interval: higher development index, high development index, medium development index, low development index,

and lower development index. According to the development index range of each type of development level, 25 ethnic minority development indexes were classified.

## 3. Results and Accuracy Verification

### 3.1. Ethnic Minority Development Index

In order to better reflect the development index of ethnic minorities, this article selecteed the Yi, Wa, Zhuang, Naxi, Hani, and Dai, six ethnic minorities with higher development indexes, for cartographic analysis. By comparing the development index calculated by Equation (3), and reference [49], the natural fracture method can most appropriately group similar values and maximize the difference between each class, so we compared the three methods of using the natural breaks method, average classification method, and manual breaks method, and found that the method using natural breaks method worked the best. This article divided the development index into five categories. The first category indicates areas with extremely poor development of the ethnic minorities, which are directly regarded as areas without the distribution of ethnic minorities. The second category indicates areas with poor development of the ethnic minority. The third category indicates areas with a moderate development. The fourth category indicates areas where the ethnic minority has developed well. The fifth category indicates areas with excellent development of the ethnic minority. The results are shown in Figures 4–9.

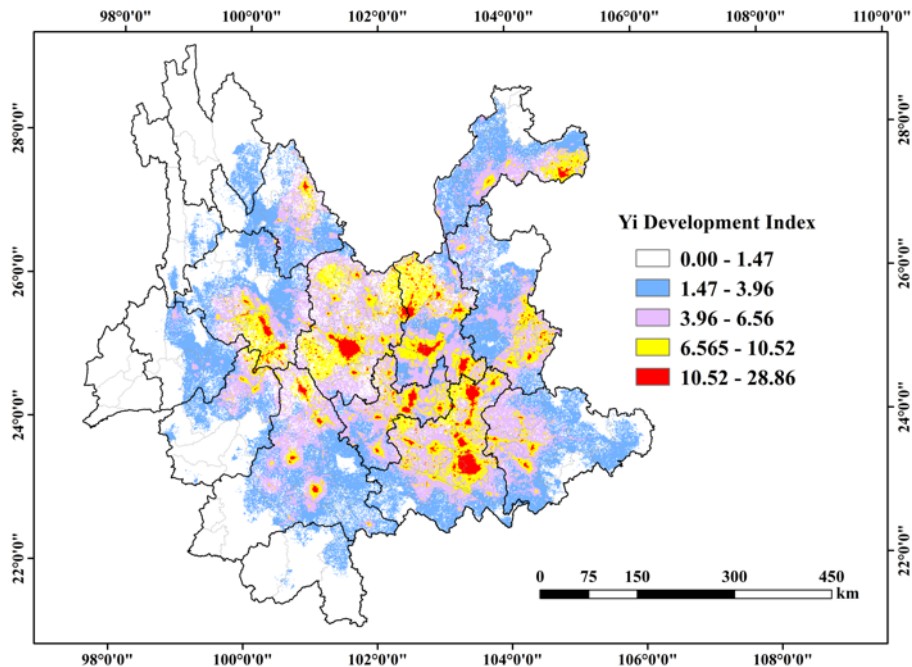

**Figure 4.** Yi nationality development index classification results.

It can be seen from Figures 4–9 that among the six ethnic minorities with high development indexes in Yunnan Province, the development indexes from high to low were: Yi, Wa, Zhuang, Naxi, Hani, and Dai. Moreover, the development index of Hani and Dai, Zhuang and Naxi were not much different. In other words, among the six ethnic minorities, the Yi ethnic group had the best development (the Yi ethnic group had the highest development index, and there were many areas with high development indexes), and the Dai ethnic group had the worst development compared to the other five ethnic minorities.

It can be seen from Figure 4 that the Yi nationality was distributed almost throughout Yunnan Province. The areas with higher Yi development index were: (1) the northeast area of Nanjian Yi County in Dali Bai Prefecture and the east area of Weishan Yi Hui County; (2) the eastern part of Chuxiong City, Chuxiong Yi Prefecture; (3) the junction of Wuhua District, Xishan District, Guandu District, and Panlong District of Kunming City,

the western part of Shilin Yi County, and the southern part of Luquan Yi and Miao County; (4) the southeast area of Eshan Yi County, Yuxi City, and the east area of Yuxi City; and (5) the northern area of Mile County, the western area of Kaiyuan City, the western area of Mengzi County, and the eastern area of Gejiu City in Honghe Hani and Yi Prefecture.

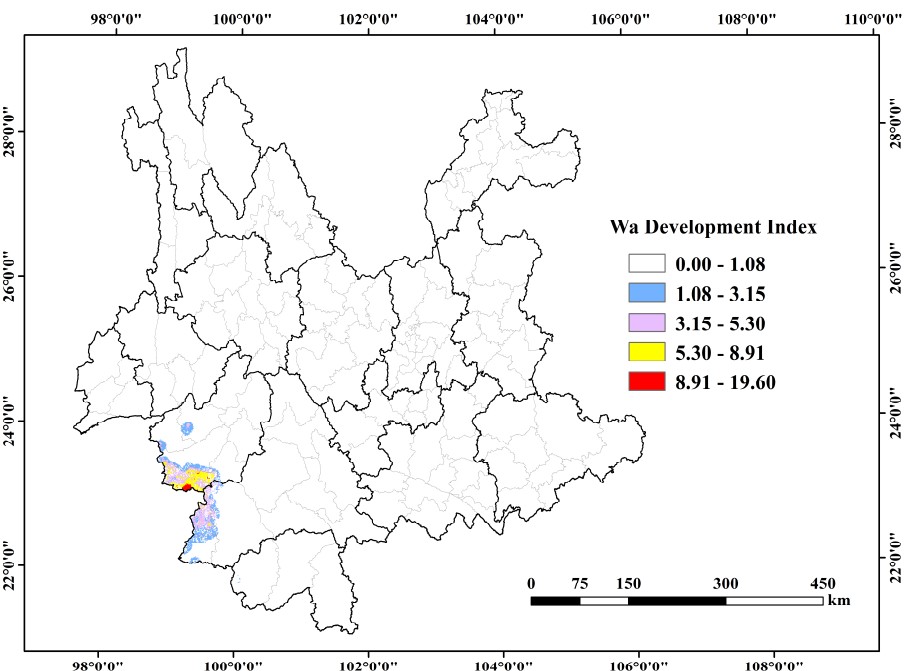

**Figure 5.** Wa nationality development index classification results.

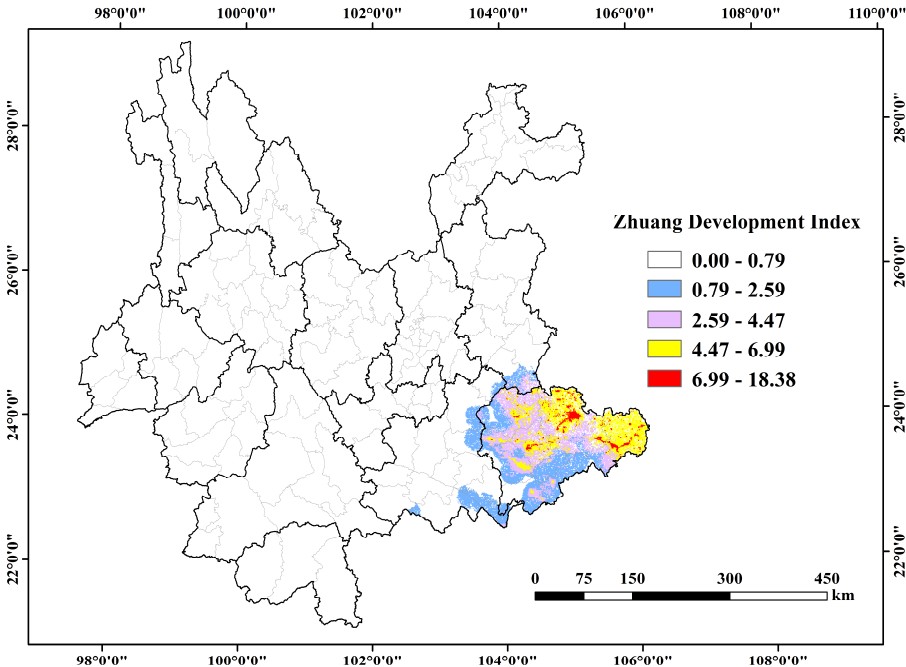

**Figure 6.** Zhuang nationality development index classification results.

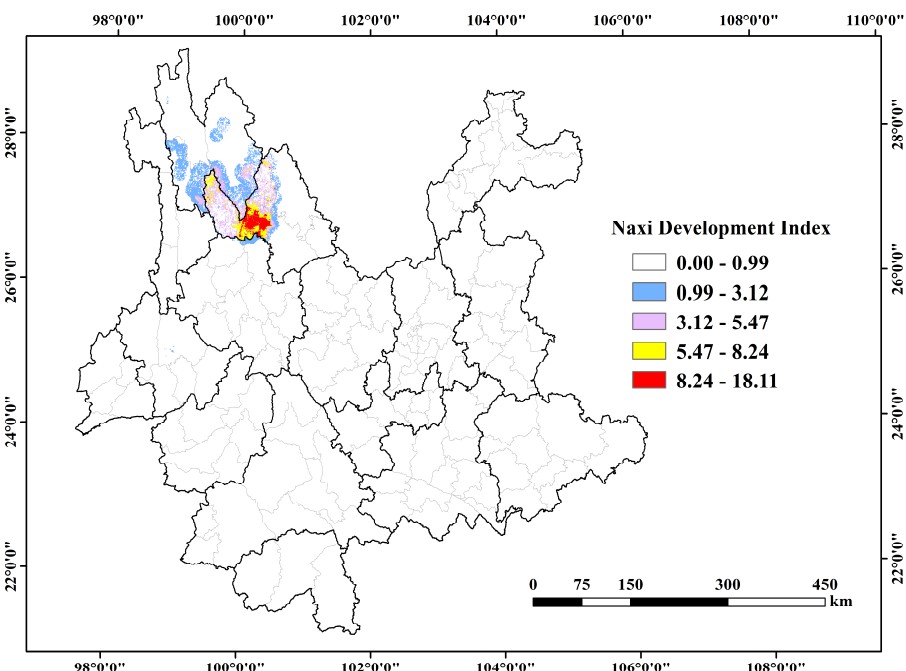

**Figure 7.** Naxi nationality development index classification results.

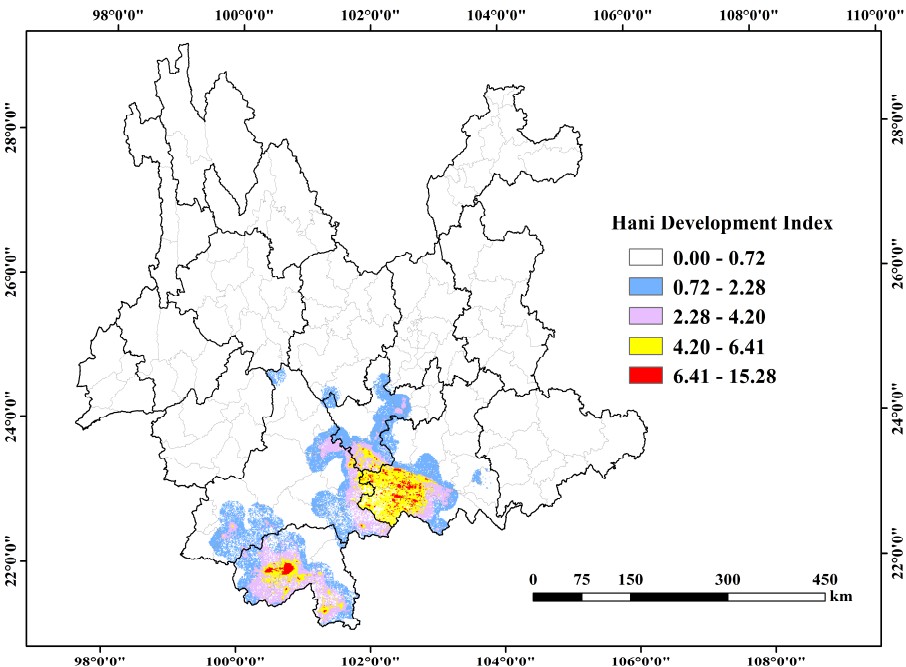

**Figure 8.** Hani nationality development index classification results.

It can be seen from Figure 5 that the distribution of the Wa nationality had regional characteristics, mainly in Cangyuan Wa County in Lincang City and Ximeng Wa County in Pu'er City. Between them, the Wa development index was the highest in the southern area of Cangyuan Wa County.

It can be seen from Figure 6 that the Zhuang nationality was mainly distributed in Wenshan Zhuang and Miao Prefecture. The areas with higher Zhuang development index were: (1) Qiubei County and the central area of Yanshan County; (2) the northern part of Funing County; and (3) the northwestern part of Guangnan County.

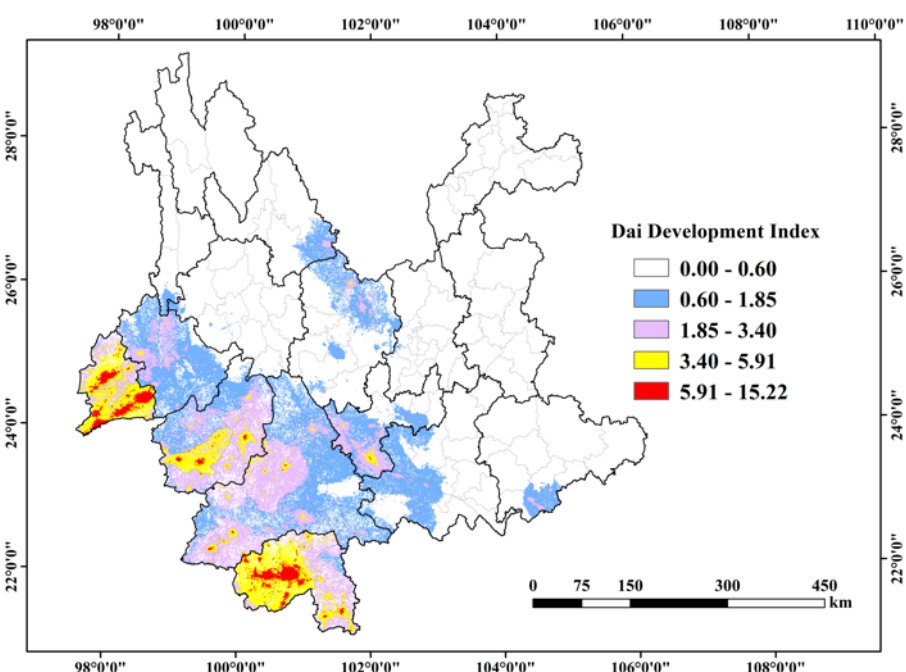

**Figure 9.** Dai nationality development index classification results.

It can be seen from Figure 7 that the Naxi nationality was mainly distributed in the western region of Lijiang City. The development index of the Naxi nationality was higher in the southern area of Lijiang urban and the southern area of Yulong Naxi County.

It can be seen from Figure 8 that the Hani nationality was mainly distributed in Xishuangbanna Dai Prefecture, southwest of Honghe Hani and Yi Prefecture, and southeast of Pu'er City. The areas with higher Hani development index were: (1) the western area of Jinghong City and the eastern area of Menghai County; (2) the central area of Hani and Yi County in Jiangyu; and (3) Honghe County, Yuanyang County, and Luchun County. It has the characteristics of not being concentrated and more scattered.

It can be seen from Figure 9 that the Dai nationality was mainly distributed in Dehong Dai Jingpo Prefecture, Lincang City, Xishuangbanna Dai Prefecture, Pu'er City, Baoshan City, and the western area of Yuxi City. The areas with higher Dai development index were: (1) Yingjiang County and Ruili City's southern area, and Mang City's central area; (2) the central area of Menghai County and Jinghong City; (3) Lincang city center and the central area of Gengma Dai and Wa County; (4) the central area of Yuanjiang County; and (5) the central areas of Menglian County, Lancang County, and Jinggu County.

*3.2. Ethnic Minority Distribution Results*

We used the method of in Section 2.2.3 to obtain the distribution results of 25 ethnic minorities in Yunnan Province (Figure 10).

It can be seen from Figure 10 that the coverage of the Yi ethnic group was the widest, involved the most counties, and was concentrated in Chuxiong Prefecture, the southeastern area of Qujing City, and the northern area of Kunming. The Jingpo branch is mainly distributed in Dehong Prefecture and Gongshan County. Zhuang Dai language branch was mainly distributed in Wenshan Prefecture, Dehong Prefecture, Xishuangbanna Prefecture, and Lincang Prefecture. The Chinese branch was mainly distributed in Zhaotong City and Baoshan City. Other language branches were mainly distributed in the east of Zhaotong and the north of Zhaotong, the east of Wenshan Prefecture, Nujiang Prefecture, Lijiang City, Dehong Prefecture, Xishuangbanna Prefecture, and the south of Lincang Prefecture.

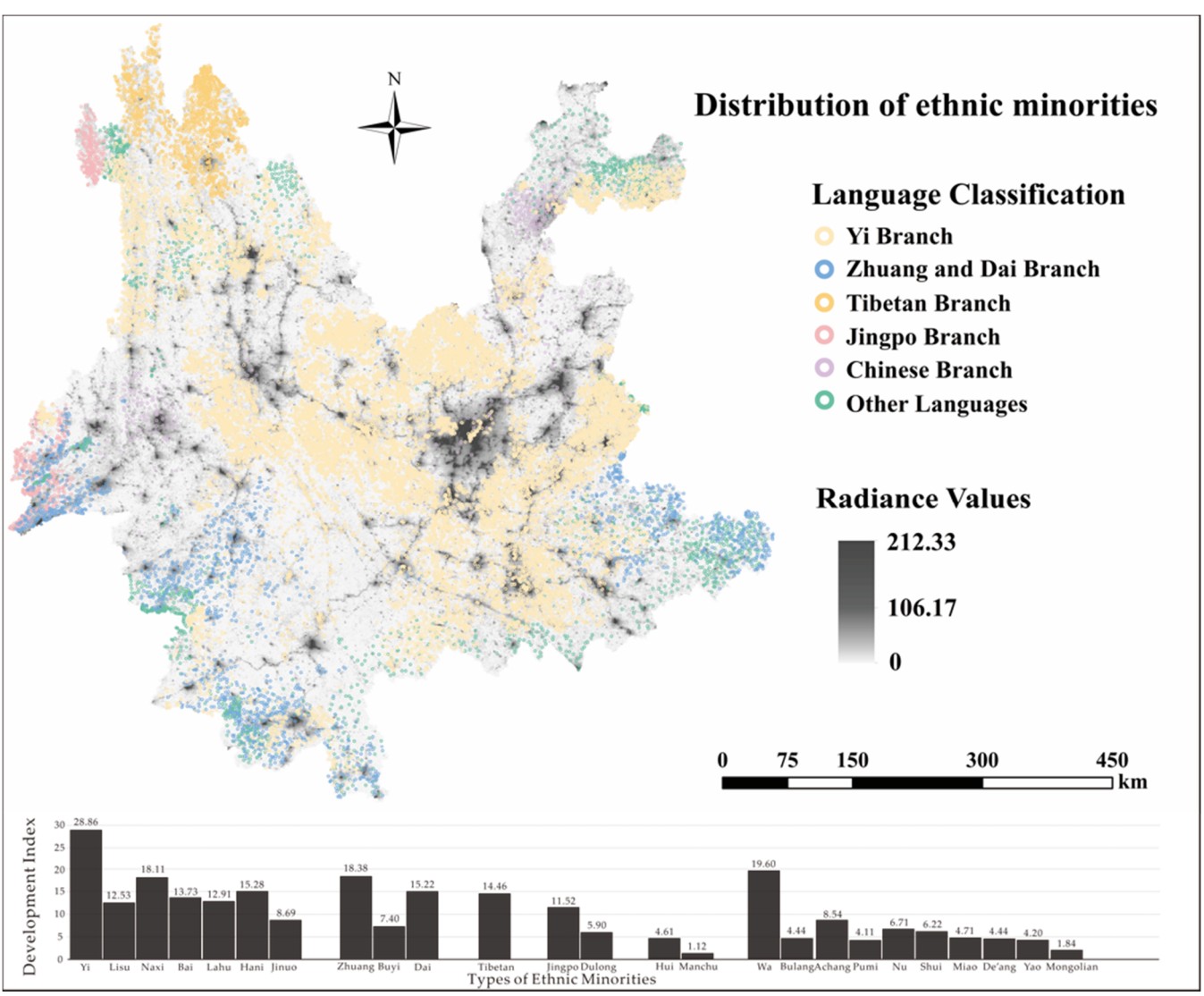

**Figure 10.** Distribution of ethnic minorities in Yunnan Province.

From the perspective of development index, the higher developed ethnic minority was the Yi. The high-developed index ethnic minorities included the Naxi, Zhuang, and Wa. The medium-developed ethnic minorities included the Hani, Bai, Lahu, Lisu, Dai, and Tibetan. The low-developed ethnic minorities included the Jingpo, Jinuo, Buyi, Achang, Nu, and Shui. The lower-developed ethnic minorities included Dulong, Bulang, Pumi, Miao, Hui, Manchu, De'ang, Yao, and Mongolian.

From the perspective of the language branch, the overall development index of ethnic minorities in the Yi, Zhuang, Dai, and Tibetan branch was relatively high, which may be related to the inheritance and development of these language branches.

Several ethnic minorities lived together in most areas. Among them, the mixed living of ethnic minorities in Dehong Prefecture was more obvious. Areas where the phenomenon of multi-ethnic mixed living was more obvious were: (1) Zhaotong City has mixed living of Yi branch, Chinese Branch, and other languages; (2) Funing County and Guangnan County in Wenshan Prefecture had mixed living of the Zhuang and Dai branch, and other languages; (3) Fumin County in Kunming City had mixed living of the Yi branch, Chinese Branch, and other languages; (4) Mengla County in Xishuangbanna Prefecture had mixed living of the Zhuang and Dai branches and other languages. (5) Jinghong City in Xishuangbanna Prefecture has mixed living of Zhuang and Dai branches and other languages, Yi branch, and Jingpo branch; (6) Longchuan County in Dehong Prefecture

had mixed living of Zhuang and Dai branches, other languages, and Jingpo branch; (7) Yingjiang County in Dehong Prefecture had mixed living of Zhuang and Dai branches, other languages, Yi branch, and Jingpo branch; and (8) Lushui County in Nujiang Prefecture had mixed living of other languages and the Yi branch.

There are currently 25 ethnic minorities in Yunnan Province, among which the Yi nationality is the most widely distributed and relatively scattered. Among the six selected ethnic minorities with the highest development index, from the perspective of each development index, the Yi nationality's development index was the highest, reaching 28.86 (to two decimal places). The Wa nationality had the second development index, reaching 19.60, but was far from the Yi nationality, which had the highest ranked development index. The Zhuang nationality had the third development index, reaching 18.38. The Naxi nationality had the fourth development index, reaching 18.11. The Hani nationality had the fifth development index, reaching 15.28. The Dai nationality's development index was the lowest at 15.22.

From the perspective of the relationship between the development index of each ethnic group and the geographic location of the ethnic group: the six areas with higher development indexes of ethnic minorities were located in the corresponding ethnic minority states, counties, or the city center of each city. Ethnic minorities had the highest development index in their corresponding minority prefecture or county, and the further the distance from the minority prefecture or county, the smaller the development index. The minority development index decreased as the distance between the minority nationality and its core development zone increased.

From the perspective of the relationship between each ethnic development index and the corresponding ethnic minority prefecture and county, areas with a higher ethnic development index were concentrated in the ethnic minority prefecture or county, but the development of the ethnic minority in the prefecture was better than that in the county.

### 3.3. Accuracy Verification

In order to verify the correctness of the development index calculated by the method used in this article, the method used in this article was compared with the method of the traditional research statistical yearbook. Considering that the development of a region is affected by many factors such as rural population, urban population, employment rate, average resident salary, etc., it is difficult to verify the correctness of the results of this article by selecting only one indicator. Comprehensively referenced in [51–53], this article selected eight indicators for the study area from 2013 to 2018. These were the total output value of agriculture, forestry, animal husbandry, and fishery in each county, and the per capita disposable income of rural residents in each county. County GDP per capita, GDP index of each county, rural employees in each county, rural population in each county, average salary of employees in each county, and number of employees in each county. Since the magnitudes of the eight indicators were different, the indicators were normalized first. The normalized formula is shown in Equation (4):

$$X = \frac{x - \min}{\max - \min},\qquad(4)$$

where $X$ is the standardized result of the index; x is the original value of the index; max is the maximum value of the sample data; and min is the minimum value of the sample data.

After obtaining the normalized results of the indicators, a comprehensive development index was established according to the method of establishing a comprehensive poverty index in the literature [53,54]. First, the entropy method was used to determine the weight of the eight indicators. In the entropy method, the larger the amount of information, the smaller the uncertainty of the information, and the smaller the entropy value, so the greater the weight. Using the entropy method to calculate the weight of each indicator, we

can obtain the comprehensive development index. The calculation formula is shown in Equations (5)–(8):

$$Z = \sum_{i=1}^{n} w_j * X_i,$$
(5)

$$f_{ij} = \frac{y_{ij}}{\sum\limits_{i=1}^{m} y_{ij}},$$
(6)

$$H_j = -(1/\ln m) \sum_{i=1}^{m} f_{ij} \ln f_{ij},$$
(7)

$$w_j = \frac{(1 - H_j)}{\sum\limits_{j=1}^{n} (1 - H_j)},$$
(8)

where $f_{ij}$ is the index value weight of the $i$ evaluation object under the $j$ index; $m$ is the 129 counties included in the study area; $n$ is the eight indicators to construct the comprehensive development index; $Z$ is the comprehensive development index; $X_i$ is the standardized result of $i$ evaluation object; $H_j$ is the entropy value of the $j$ index; and $w_j$ is the weight of the $j$ index.

The weights of the eight indicators in 2013–2018 calculated by the formula are shown in Table 3 (with four decimal places).

**Table 3.** The calculation results of the weight of each indicator from 2013 to 2018.

| Indicators \ Year | 2013 | 2014 | 2015 | 2016 | 2017 | 2018 |
|---|---|---|---|---|---|---|
| GDP per capita | 0.1446 | 0.1401 | 0.1350 | 0.1304 | 0.1321 | 0.1330 |
| GDP Index | 0.0419 | 0.0152 | 0.0252 | 0.0183 | 0.0144 | 0.0277 |
| Number of employees | 0.3597 | 0.3630 | 0.3752 | 0.3732 | 0.3755 | 0.3716 |
| Average salary of employees | 0.0057 | 0.0059 | 0.0052 | 0.0086 | 0.0090 | 0.0093 |
| Per capita disposable income of rural residents | 0.0171 | 0.0239 | 0.0232 | 0.0230 | 0.0225 | 0.0222 |
| Total output value of agriculture, forestry, animal husbandry and fishery | 0.1569 | 0.1503 | 0.1456 | 0.1453 | 0.1444 | 0.1438 |
| Rural population | 0.1636 | 0.1646 | 0.1649 | 0.1667 | 0.1641 | 0.1665 |
| Rural workers | 0.1520 | 0.1507 | 0.1507 | 0.1526 | 0.1523 | 0.1533 |

According to the calculation results of the weight of each index, the comprehensive development index of each county in the study area was obtained, as shown in Table 4.

**Table 4.** The results of the comprehensive development index of each county from 2013 to 2018.

| County \ Year | 2013 | 2014 | 2015 | 2016 | 2017 | 2018 |
|---|---|---|---|---|---|---|
| Wuhua | 0.5565 | 0.4184 | 0.4553 | 0.5595 | 0.5919 | 0.5626 |
| Panlong | 0.4935 | 0.5719 | 0.4175 | 0.5943 | 0.4999 | 0.5973 |
| Guandu | 0.4058 | 0.5501 | 0.5886 | 0.5818 | 0.5571 | 0.5577 |
| Xishan | 0.4362 | 0.5920 | 0.3979 | 0.3641 | 0.5711 | 0.3423 |
| Dongchuan | 0.2069 | 0.1855 | 0.1970 | 0.1763 | 0.1263 | 0.1595 |
| Chenggong | 0.1706 | 0.2412 | 0.3289 | 0.5510 | 0.5790 | 0.5906 |
| Jinning | 0.2200 | 0.1880 | 0.1278 | 0.1258 | 0.1740 | 0.1603 |
| Fumin | 0.0309 | 0.1454 | 0.1275 | 0.0793 | 0.0285 | 0.1881 |
| Yiliang | 0.0802 | 0.1650 | 0.1813 | 0.2292 | 0.2224 | 0.2432 |
| Shilin | 0.1635 | 0.3860 | 0.0723 | 0.1983 | 0.1256 | 0.1756 |
| Songming | 0.2307 | 0.1250 | 0.1700 | 0.1531 | 0.2550 | 0.2424 |
| Luquan | 0.1190 | 0.0354 | 0.0404 | 0.2198 | 0.0635 | 0.0625 |
| Xundian | 0.0482 | 0.0228 | 0.1243 | 0.1005 | 0.0759 | 0.0975 |
| Anning | 0.1812 | 0.0925 | 0.1538 | 0.2232 | 0.2606 | 0.2492 |

**Table 4.** *Cont.*

| County \ Year | 2013 | 2014 | 2015 | 2016 | 2017 | 2018 |
|---|---|---|---|---|---|---|
| Qilin | 0.2941 | 0.2193 | 0.2126 | 0.3072 | 0.2386 | 0.2395 |
| Malong | 0.1123 | 0.1928 | 0.1693 | 0.0839 | 0.1293 | 0.2136 |
| Luliang | 0.1289 | 0.1170 | 0.0310 | 0.0507 | 0.1856 | 0.0680 |
| Shizong | 0.0929 | 0.0283 | 0.1104 | 0.0220 | 0.0602 | 0.1154 |
| Luoping | 0.0533 | 0.0200 | 0.1049 | 0.1091 | 0.1217 | 0.1467 |
| Fuyuan | 0.1455 | 0.1421 | 0.1275 | 0.1070 | 0.1327 | 0.1951 |
| Huize | 0.1337 | 0.1262 | 0.0275 | 0.0728 | 0.0610 | 0.0458 |
| Zhanyi | 0.1868 | 0.1564 | 0.1065 | 0.0506 | 0.0582 | 0.1565 |
| Xuanwei | 0.1228 | 0.1014 | 0.1931 | 0.0926 | 0.1288 | 0.1064 |
| Hongta | 0.3898 | 0.2574 | 0.2017 | 0.2177 | 0.3029 | 0.2844 |
| Jiangchuan | 0.2124 | 0.2147 | 0.1954 | 0.1948 | 0.2097 | 0.2263 |
| Chengjiang | 0.2890 | 0.2553 | 0.2037 | 0.1633 | 0.2272 | 0.2198 |
| Tonghai | 0.1425 | 0.0818 | 0.1842 | 0.2408 | 0.2071 | 0.2459 |
| Huaning | 0.0806 | 0.0789 | 0.1427 | 0.0529 | 0.1255 | 0.1587 |
| Yimen | 0.0270 | 0.1556 | 0.1092 | 0.0388 | 0.0239 | 0.1186 |
| Eshan | 0.1073 | 0.1150 | 0.0641 | 0.1003 | 0.0377 | 0.1089 |
| Xinping | 0.0507 | 0.1436 | 0.0201 | 0.0203 | 0.1204 | 0.1154 |
| Yuanjiang | 0.0563 | 0.0205 | 0.0623 | 0.0751 | 0.0713 | 0.1310 |
| Longyang | 0.0422 | 0.0615 | 0.1098 | 0.0220 | 0.0541 | 0.0646 |
| Shidian | 0.0196 | 0.0898 | 0.0342 | 0.0203 | 0.0542 | 0.1068 |
| Tengchong | 0.0290 | 0.0575 | 0.0766 | 0.0439 | 0.0550 | 0.1588 |
| Longling | 0.1197 | 0.0751 | 0.0888 | 0.0423 | 0.0244 | 0.0209 |
| Changning | 0.0450 | 0.0403 | 0.0283 | 0.0782 | 0.0783 | 0.1131 |
| Zhaoyang | 0.1454 | 0.0551 | 0.1440 | 0.1380 | 0.1399 | 0.1832 |
| Ludian | 0.0938 | 0.1802 | 0.1376 | 0.1312 | 0.1831 | 0.2062 |
| Qiaojia | 0.1017 | 0.0723 | 0.0385 | 0.0712 | 0.0211 | 0.1004 |
| Yanjin | 0.0391 | 0.0204 | 0.0759 | 0.0422 | 0.1075 | 0.1131 |
| Daguan | 0.1040 | 0.0895 | 0.0613 | 0.1594 | 0.1380 | 0.1172 |
| Yongshan | 0.1071 | 0.1135 | 0.0575 | 0.1412 | 0.0964 | 0.0843 |
| Suijiang | 0.1458 | 0.0706 | 0.0896 | 0.0864 | 0.0877 | 0.0756 |
| Zhenxiong | 0.0600 | 0.0408 | 0.0878 | 0.0424 | 0.0297 | 0.0558 |
| Yiliang | 0.1195 | 0.1198 | 0.1100 | 0.0821 | 0.0885 | 0.1049 |
| Weixin | 0.0859 | 0.0209 | 0.0809 | 0.1667 | 0.1223 | 0.0757 |
| Shuifu | 0.0223 | 0.0302 | 0.0634 | 0.0380 | 0.0706 | 0.1554 |
| Gucheng | 0.0924 | 0.0613 | 0.0887 | 0.0946 | 0.1542 | 0.1332 |
| Yulong | 0.0197 | 0.0201 | 0.0578 | 0.0211 | 0.0302 | 0.0342 |
| Yongsheng | 0.0447 | 0.0215 | 0.0626 | 0.0393 | 0.0251 | 0.0568 |
| Huaping | 0.0271 | 0.0917 | 0.0322 | 0.0201 | 0.0556 | 0.0243 |
| Ninglang | 0.0247 | 0.0216 | 0.0632 | 0.0393 | 0.1433 | 0.0790 |
| Simao | 0.0651 | 0.0913 | 0.0317 | 0.0801 | 0.0302 | 0.0564 |
| Ning'er | 0.0214 | 0.0308 | 0.0895 | 0.0216 | 0.0480 | 0.1134 |
| Mojiang | 0.0253 | 0.0327 | 0.0297 | 0.0344 | 0.0605 | 0.0199 |
| Jingdong | 0.0474 | 0.1037 | 0.0204 | 0.1062 | 0.1195 | 0.0205 |
| Jinggu | 0.1055 | 0.1084 | 0.0374 | 0.0449 | 0.0464 | 0.0528 |
| Zhenyuan | 0.0360 | 0.0397 | 0.0409 | 0.0687 | 0.0601 | 0.0623 |
| Jiangcheng | 0.0264 | 0.0302 | 0.0202 | 0.0435 | 0.1008 | 0.0237 |
| Menglian | 0.0294 | 0.0202 | 0.0457 | 0.0207 | 0.0453 | 0.0267 |
| Lancang | 0.0671 | 0.0458 | 0.0204 | 0.0510 | 0.0640 | 0.1807 |
| Ximeng | 0.0813 | 0.0196 | 0.0208 | 0.0767 | 0.0207 | 0.2507 |
| Linxiang | 0.0295 | 0.0204 | 0.1297 | 0.0450 | 0.0535 | 0.0206 |
| Fengqing | 0.0199 | 0.1145 | 0.0513 | 0.0888 | 0.0207 | 0.1274 |
| Yunxian | 0.0334 | 0.0656 | 0.0211 | 0.0209 | 0.0321 | 0.0962 |
| Yongde | 0.0273 | 0.0264 | 0.0206 | 0.0403 | 0.1061 | 0.1245 |
| Zhenkang | 0.1196 | 0.0242 | 0.1059 | 0.0640 | 0.0198 | 0.0520 |
| Shuangjiang | 0.0833 | 0.0486 | 0.0206 | 0.0440 | 0.0528 | 0.0883 |
| Gengma | 0.0389 | 0.0459 | 0.0480 | 0.0325 | 0.0205 | 0.0563 |
| Cangyuan | 0.0317 | 0.0201 | 0.0312 | 0.0276 | 0.0352 | 0.0569 |
| Chuxiong | 0.0251 | 0.0231 | 0.1089 | 0.0338 | 0.2816 | 0.1152 |

**Table 4.** *Cont.*

| County \ Year | 2013 | 2014 | 2015 | 2016 | 2017 | 2018 |
|---|---|---|---|---|---|---|
| Shuangbo | 0.0299 | 0.0536 | 0.0210 | 0.1057 | 0.0997 | 0.0580 |
| Mouding | 0.1196 | 0.0204 | 0.1000 | 0.1567 | 0.2166 | 0.1957 |
| Nanhua | 0.0581 | 0.0204 | 0.1086 | 0.0968 | 0.1148 | 0.1205 |
| Yao'an | 0.0496 | 0.0685 | 0.0392 | 0.0620 | 0.0336 | 0.0608 |
| Dayao | 0.0471 | 0.0208 | 0.0578 | 0.0443 | 0.0625 | 0.0219 |
| Yongren | 0.0384 | 0.0339 | 0.0499 | 0.0897 | 0.0200 | 0.0800 |
| Yuanmou | 0.0200 | 0.0208 | 0.0824 | 0.0914 | 0.0456 | 0.0410 |
| Wuding | 0.0621 | 0.0281 | 0.0825 | 0.0366 | 0.0553 | 0.2346 |
| Lufeng | 0.0287 | 0.0322 | 0.0937 | 0.0561 | 0.1170 | 0.1709 |
| Mengzi | 0.0761 | 0.0203 | 0.0214 | 0.0760 | 0.1079 | 0.0890 |
| Gejiu | 0.0284 | 0.1278 | 0.1015 | 0.0870 | 0.1264 | 0.1201 |
| Kaiyuan | 0.0455 | 0.1137 | 0.1624 | 0.1265 | 0.1662 | 0.1792 |
| Mile | 0.0906 | 0.1955 | 0.1257 | 0.1705 | 0.1895 | 0.1721 |
| Pingbian | 0.0353 | 0.0507 | 0.0206 | 0.0807 | 0.1048 | 0.0269 |
| Jianshui | 0.0201 | 0.0705 | 0.0245 | 0.0418 | 0.1063 | 0.0564 |
| Shiping | 0.0199 | 0.0380 | 0.1395 | 0.0611 | 0.1143 | 0.0946 |
| Luxi | 0.1447 | 0.0652 | 0.0994 | 0.0891 | 0.0717 | 0.0992 |
| Yuanyang | 0.0199 | 0.0638 | 0.0708 | 0.0309 | 0.0922 | 0.0706 |
| Honghe | 0.0230 | 0.1195 | 0.0217 | 0.0256 | 0.0238 | 0.1657 |
| Jinping | 0.0509 | 0.0202 | 0.0208 | 0.0207 | 0.0317 | 0.0206 |
| Luchun | 0.0477 | 0.0621 | 0.0290 | 0.0359 | 0.0618 | 0.0369 |
| Hekou | 0.0678 | 0.0488 | 0.0540 | 0.0390 | 0.0466 | 0.0206 |
| Wenshan | 0.1196 | 0.0734 | 0.0784 | 0.0527 | 0.1758 | 0.1454 |
| Yanshan | 0.0590 | 0.0655 | 0.0204 | 0.0804 | 0.0204 | 0.1157 |
| Xichou | 0.1082 | 0.0495 | 0.0737 | 0.0613 | 0.0949 | 0.1313 |
| Malipo | 0.1195 | 0.1039 | 0.0898 | 0.0211 | 0.1296 | 0.2052 |
| Maguan | 0.0538 | 0.0537 | 0.0206 | 0.0208 | 0.0878 | 0.1211 |
| Qiubei | 0.0212 | 0.0522 | 0.0347 | 0.0264 | 0.0746 | 0.0210 |
| Guangnan | 0.1195 | 0.0275 | 0.0352 | 0.0208 | 0.0579 | 0.0204 |
| Funing | 0.0200 | 0.0202 | 0.0390 | 0.0370 | 0.0244 | 0.0225 |
| Jinghong | 0.0339 | 0.0985 | 0.1048 | 0.1219 | 0.1242 | 0.0996 |
| Menghai | 0.0286 | 0.0216 | 0.0309 | 0.0324 | 0.0738 | 0.0533 |
| Mengla | 0.0546 | 0.1012 | 0.1310 | 0.1316 | 0.0516 | 0.1163 |
| Dali | 0.2015 | 0.1204 | 0.2399 | 0.2192 | 0.1301 | 0.1922 |
| Yangbi | 0.0202 | 0.0489 | 0.0407 | 0.0577 | 0.1129 | 0.0841 |
| Xiangyun | 0.0971 | 0.0307 | 0.0752 | 0.0860 | 0.1438 | 0.1869 |
| Binchuan | 0.0358 | 0.1710 | 0.0278 | 0.0977 | 0.1210 | 0.0878 |
| Midu | 0.0577 | 0.1017 | 0.0822 | 0.1511 | 0.0946 | 0.1239 |
| Nanjian | 0.0218 | 0.0967 | 0.0246 | 0.0510 | 0.1526 | 0.0868 |
| Weishan | 0.0534 | 0.1220 | 0.0705 | 0.1026 | 0.0599 | 0.0919 |
| Yongping | 0.0740 | 0.0848 | 0.0351 | 0.0952 | 0.0422 | 0.0622 |
| Yunlong | 0.0422 | 0.1352 | 0.0734 | 0.1527 | 0.1387 | 0.2176 |
| Eryuan | 0.0214 | 0.0419 | 0.0310 | 0.0206 | 0.0202 | 0.1260 |
| Jianchuan | 0.0379 | 0.0366 | 0.0211 | 0.0832 | 0.1196 | 0.0851 |
| Heqing | 0.0715 | 0.0213 | 0.0658 | 0.0636 | 0.0688 | 0.0851 |
| Mangshi | 0.0200 | 0.1012 | 0.1463 | 0.1428 | 0.1542 | 0.0607 |
| Ruili | 0.0626 | 0.1126 | 0.0557 | 0.0518 | 0.1487 | 0.2090 |
| Lianghe | 0.0343 | 0.1255 | 0.0221 | 0.0202 | 0.1423 | 0.1570 |
| Yingjiang | 0.1195 | 0.0694 | 0.1095 | 0.0374 | 0.2701 | 0.0613 |
| Longchuan | 0.0256 | 0.0949 | 0.0204 | 0.0198 | 0.0495 | 0.1184 |
| Lushui | 0.0209 | 0.0219 | 0.0874 | 0.1034 | 0.0254 | 0.0943 |
| Fugong | 0.0289 | 0.0288 | 0.0217 | 0.0427 | 0.0465 | 0.0644 |
| Gongshan | 0.0204 | 0.0498 | 0.0332 | 0.1214 | 0.0707 | 0.0924 |
| Lanping | 0.0272 | 0.0383 | 0.0282 | 0.0199 | 0.0686 | 0.0912 |
| Shangri-La | 0.3128 | 0.4385 | 0.2572 | 0.4639 | 0.4023 | 0.4179 |
| Deqin | 0.0506 | 0.0309 | 0.0282 | 0.0950 | 0.0798 | 0.0859 |
| Weixi | 0.0772 | 0.0261 | 0.0730 | 0.1198 | 0.0490 | 0.0460 |

Using the method developed in this article to calculate the comprehensive development index of all ethnic minorities and Han nationality in the study area from 2013 to 2018, we performed district statistics on the development index of each county on ArcMap, and took the average value of the development index of each county as the statistical value. The development index of each county from 2013 to 2018 is shown in Table 5.

**Table 5.** The development index result calculated by the method in this paper.

| County \ Year | 2013 | 2014 | 2015 | 2016 | 2017 | 2018 |
|---|---|---|---|---|---|---|
| Wuhua | 2333.1825 | 2339.1373 | 2376.7834 | 2394.4854 | 2415.8620 | 2448.1288 |
| Panlong | 2470.0052 | 2554.7521 | 2333.1047 | 2404.9305 | 2504.0174 | 2511.2501 |
| Guandu | 2170.9487 | 2259.7741 | 2353.8594 | 2381.0156 | 2449.4958 | 2561.8397 |
| Xishan | 1621.7904 | 1784.9426 | 1598.3018 | 1565.5310 | 1597.4138 | 1602.9692 |
| Dongchuan | 960.6572 | 994.8467 | 742.9193 | 836.3549 | 854.5164 | 934.8275 |
| Chenggong | 943.5753 | 1076.5617 | 1205.3264 | 2298.8631 | 2363.8025 | 2420.8939 |
| Jinning | 595.8789 | 627.4076 | 688.4572 | 711.8920 | 830.7505 | 850.9899 |
| Fumin | 612.8931 | 695.4732 | 746.5198 | 784.6463 | 789.6833 | 897.2893 |
| Yiliang | 625.2049 | 761.7744 | 830.1322 | 886.5681 | 962.0898 | 996.5625 |
| Shilin | 766.8347 | 789.6061 | 794.6279 | 825.1524 | 891.7881 | 985.6760 |
| Songming | 899.6777 | 927.3485 | 946.6784 | 973.7795 | 1107.7790 | 1056.7363 |
| Luquan | 320.2183 | 364.9809 | 400.2450 | 438.9507 | 448.5508 | 576.7864 |
| Xundian | 517.4374 | 522.8907 | 539.6836 | 574.8628 | 601.3995 | 746.0349 |
| Anning | 945.3103 | 955.8508 | 957.5187 | 1008.2649 | 1050.5527 | 1084.9657 |
| Qilin | 1069.2507 | 1084.1809 | 1090.9449 | 1116.4519 | 1172.8232 | 1240.0189 |
| Malong | 649.2351 | 693.1274 | 712.2659 | 729.2838 | 757.7143 | 860.7674 |
| Luliang | 543.4304 | 554.4877 | 576.5027 | 589.5539 | 595.1228 | 638.4553 |
| Shizong | 483.0773 | 485.1854 | 500.8513 | 519.5551 | 526.3722 | 657.7111 |
| Luoping | 491.1515 | 417.1506 | 518.5569 | 531.6147 | 554.2872 | 672.6678 |
| Fuyuan | 633.7029 | 641.5798 | 652.7819 | 663.3065 | 674.2953 | 782.7485 |
| Huize | 617.1130 | 647.3857 | 481.3886 | 507.0445 | 558.0216 | 605.9378 |
| Zhanyi | 600.3138 | 635.8879 | 660.7369 | 673.4620 | 696.3811 | 843.4202 |
| Xuanwei | 623.7029 | 656.5798 | 662.7819 | 673.3065 | 684.2953 | 796.7485 |
| Hongta | 1276.6486 | 1310.7777 | 1255.3083 | 1207.8354 | 1358.3035 | 1377.4974 |
| Jiangchuan | 897.7480 | 912.4791 | 947.5681 | 952.9043 | 1104.6690 | 1133.9384 |
| Chengjiang | 1004.5388 | 1031.805437 | 1047.223082 | 1074.157544 | 1149.89266 | 1157.818511 |
| Tonghai | 883.0552 | 919.5813 | 957.5814 | 990.6898 | 1062.5763 | 1117.2477 |
| Huaning | 585.7714 | 595.2825 | 640.7928 | 658.2505 | 715.3348 | 811.3422 |
| Yimen | 494.4001 | 511.3989 | 532.8776 | 551.4176 | 555.8165 | 693.6572 |
| Eshan | 489.2267 | 549.9414 | 580.4157 | 593.6294 | 615.8568 | 664.5724 |
| Xinping | 260.6868 | 421.3788 | 415.0625 | 415.7112 | 453.7176 | 537.3233 |
| Yuanjiang | 486.8459 | 539.9640 | 559.4845 | 596.8433 | 624.0441 | 689.0722 |
| Longyang | 433.3167 | 507.6858 | 521.9453 | 535.2050 | 590.8690 | 664.5847 |
| Shidian | 297.5054 | 455.6399 | 417.2362 | 473.0716 | 502.3391 | 582.7547 |
| Tengchong | 381.8499 | 569.4589 | 552.5687 | 604.0656 | 669.7669 | 753.0422 |
| Longling | 283.7437 | 472.8197 | 460.4084 | 499.1058 | 498.2598 | 612.5852 |
| Changning | 299.6858 | 441.3997 | 467.5553 | 447.8070 | 469.2807 | 565.8577 |
| Zhaoyang | 803.7647 | 813.1024 | 880.7033 | 856.9120 | 885.1878 | 954.2087 |
| Ludian | 640.0504 | 697.3706 | 741.4808 | 756.5885 | 772.3398 | 868.5031 |
| Qiaojia | 461.1212 | 479.4775 | 498.7038 | 540.6888 | 592.3084 | 670.8998 |
| Yanjin | 427.3850 | 495.8935 | 540.4706 | 557.9826 | 565.5458 | 616.2725 |
| Daguan | 529.7191 | 622.5744 | 655.1364 | 689.8198 | 755.0185 | 767.3721 |
| Yongshan | 696.0273 | 684.1769 | 741.3797 | 744.1170 | 810.3069 | 884.5728 |
| Suijiang | 626.3479 | 600.9896 | 614.0467 | 641.6444 | 706.0032 | 718.7469 |
| Zhenxiong | 448.4068 | 454.1750 | 578.5055 | 497.7722 | 473.2344 | 564.0579 |
| Yiliang | 455.5056 | 426.6301 | 535.4907 | 578.2334 | 588.6284 | 642.6751 |
| Weixin | 649.2141 | 576.1327 | 623.3148 | 665.7798 | 808.1880 | 817.0238 |
| Shuifu | 477.7044 | 536.3990 | 554.4428 | 565.6362 | 624.9817 | 672.7901 |
| Gucheng | 715.5696 | 656.3005 | 791.5617 | 844.6705 | 659.9559 | 730.8966 |
| Yulong | 249.8526 | 278.7471 | 289.6164 | 322.6217 | 348.7171 | 434.7898 |
| Yongsheng | 306.8789 | 315.7225 | 339.0830 | 358.6972 | 367.6797 | 455.4456 |
| Huaping | 398.2136 | 435.1668 | 448.9292 | 465.1374 | 473.1283 | 529.3849 |

**Table 5.** *Cont.*

| County | Year 2013 | 2014 | 2015 | 2016 | 2017 | 2018 |
|---|---|---|---|---|---|---|
| Ninglang | 483.3725 | 499.1192 | 503.1318 | 532.7423 | 568.5116 | 556.7700 |
| Simao | 262.0737 | 378.4076 | 411.3063 | 423.9178 | 428.7674 | 515.5211 |
| Ning'er | 205.7700 | 345.3506 | 377.4181 | 363.9567 | 380.2126 | 492.3221 |
| Mojiang | 294.4344 | 320.0161 | 345.7805 | 360.6149 | 368.5020 | 492.5927 |
| Jingdong | 285.6620 | 323.7440 | 442.4544 | 445.8952 | 451.9286 | 557.7185 |
| Jinggu | 204.1723 | 204.5097 | 286.7719 | 300.7873 | 279.4673 | 347.8470 |
| Zhenyuan | 316.8741 | 345.1779 | 366.7534 | 369.0427 | 369.2104 | 463.5557 |
| Jiangcheng | 259.2867 | 353.5767 | 385.5516 | 390.8240 | 468.7765 | 490.3128 |
| Menglian | 214.4475 | 261.5678 | 354.3153 | 353.2323 | 388.4604 | 440.2223 |
| Lancang | 111.0612 | 131.2043 | 226.3668 | 288.6047 | 273.7476 | 333.1441 |
| Ximeng | 112.9200 | 172.0400 | 326.1849 | 337.2044 | 349.1749 | 407.7943 |
| Linxiang | 306.9635 | 462.6050 | 515.0650 | 507.7827 | 502.8663 | 588.0734 |
| Fengqing | 380.4210 | 587.8980 | 589.8904 | 608.2613 | 611.0447 | 714.5013 |
| Yunxian | 315.0365 | 499.6152 | 461.8195 | 532.8510 | 521.2354 | 619.7606 |
| Yongde | 226.1955 | 423.5674 | 434.0769 | 478.3543 | 459.3394 | 528.0003 |
| Zhenkang | 297.1114 | 355.5066 | 364.0461 | 370.0390 | 376.3777 | 433.9040 |
| Shuangjiang | 227.5567 | 287.2707 | 326.3009 | 387.4047 | 401.0221 | 468.4001 |
| Gengma | 217.7960 | 347.9124 | 396.2941 | 372.3042 | 356.5077 | 406.0964 |
| Cangyuan | 160.4840 | 309.5874 | 338.4399 | 339.9750 | 354.0869 | 416.7235 |
| Chuxiong | 433.2826 | 645.2557 | 653.4692 | 657.8736 | 681.1811 | 837.8980 |
| Shuangbo | 252.8526 | 303.8850 | 379.6348 | 390.3115 | 407.8300 | 520.8493 |
| Mouding | 310.4608 | 560.9061 | 615.7034 | 638.1917 | 696.1406 | 832.3970 |
| Nanhua | 480.4072 | 505.4462 | 609.2866 | 625.6880 | 638.3771 | 782.9123 |
| Yao'an | 209.0912 | 390.8838 | 467.7279 | 486.7033 | 523.6015 | 651.2314 |
| Dayao | 252.5988 | 349.5919 | 369.2466 | 413.5369 | 424.8047 | 538.7005 |
| Yongren | 345.9574 | 362.8734 | 373.2791 | 374.4745 | 446.5159 | 482.9805 |
| Yuanmou | 339.6947 | 476.3847 | 520.1060 | 529.2911 | 571.1880 | 670.1142 |
| Wuding | 247.2879 | 375.7019 | 384.2886 | 436.3690 | 459.5321 | 573.1000 |
| Lufeng | 478.5784 | 580.9119 | 586.9430 | 591.4193 | 655.7322 | 748.1127 |
| Mengzi | 457.9591 | 464.1473 | 470.5014 | 506.2215 | 511.0077 | 613.6647 |
| Gejiu | 572.3553 | 600.1590 | 591.4074 | 642.3799 | 690.1944 | 748.6155 |
| Kaiyuan | 668.0048 | 673.1953 | 604.3685 | 624.1396 | 766.4539 | 826.0011 |
| Mile | 671.0291 | 722.1276 | 726.0442 | 747.2656 | 792.8220 | 868.9230 |
| Pingbian | 305.4681 | 375.1979 | 417.6111 | 443.0900 | 444.3564 | 572.0882 |
| Jianshui | 399.9278 | 458.7829 | 479.0721 | 504.5566 | 511.8208 | 618.5614 |
| Shiping | 321.6694 | 424.4328 | 438.0587 | 446.9695 | 470.7730 | 563.0486 |
| Luxi | 592.6135 | 609.4124 | 631.0997 | 645.3875 | 709.2802 | 801.2388 |
| Yuanyang | 333.5136 | 363.3037 | 451.0659 | 478.7271 | 523.0667 | 617.0672 |
| Honghe | 258.5071 | 372.6793 | 447.6026 | 464.3542 | 490.1732 | 609.1840 |
| Jinping | 204.2935 | 221.5390 | 339.2894 | 356.5964 | 360.8186 | 465.1307 |
| Luchun | 91.9753 | 243.7047 | 264.9559 | 309.4107 | 322.8647 | 434.3615 |
| Hekou | 272.5834 | 350.7499 | 384.7263 | 391.4023 | 396.9577 | 475.4815 |
| Wenshan | 474.0788 | 490.4182 | 499.7623 | 533.9054 | 594.8506 | 672.3611 |
| Yanshan | 421.5922 | 437.0946 | 445.0354 | 464.4252 | 494.4376 | 596.4948 |
| Xichou | 508.0878 | 515.4572 | 575.0413 | 617.1873 | 628.6801 | 727.9660 |
| Malipo | 266.5631 | 402.8942 | 460.5780 | 461.5752 | 485.8210 | 588.7415 |
| Maguan | 293.9284 | 356.8797 | 399.5391 | 404.3396 | 448.7799 | 523.6682 |
| Qiubei | 331.6157 | 347.3116 | 384.0514 | 401.5665 | 404.9738 | 504.4938 |
| Guangnan | 266.2608 | 293.2710 | 311.7878 | 352.7943 | 359.1112 | 454.3619 |
| Funing | 240.5025 | 324.5929 | 340.5920 | 348.3029 | 363.3480 | 485.1750 |
| Jinghong | 399.9671 | 410.0248 | 452.5865 | 539.5053 | 556.8493 | 620.9858 |
| Menghai | 229.7981 | 276.6386 | 298.5195 | 306.0052 | 308.7728 | 339.3909 |
| Mengla | 562.3500 | 597.2580 | 617.6430 | 697.8365 | 700.8417 | 782.5778 |
| Dali | 894.2742 | 929.0856 | 952.3336 | 972.2778 | 1017.5021 | 1054.9061 |
| Yangbi | 409.4086 | 504.7738 | 535.1718 | 541.4113 | 592.4125 | 669.4222 |
| Xiangyun | 521.8522 | 635.7725 | 637.0776 | 639.2351 | 659.8403 | 773.7537 |
| Binchuan | 477.8487 | 515.6735 | 532.8143 | 547.7162 | 554.5976 | 662.9385 |
| Midu | 515.8058 | 614.6749 | 661.6881 | 669.8814 | 676.4856 | 796.9106 |

**Table 5.** *Cont.*

| County \ Year | 2013 | 2014 | 2015 | 2016 | 2017 | 2018 |
|---|---|---|---|---|---|---|
| Nanjian | 410.8299 | 532.0153 | 545.4751 | 554.5118 | 581.0947 | 724.9630 |
| Weishan | 495.9411 | 521.2475 | 521.6193 | 521.8876 | 559.1663 | 680.8504 |
| Yongping | 440.5821 | 478.1848 | 494.3199 | 528.8982 | 518.7397 | 631.0074 |
| Yunlong | 598.6572 | 619.4169 | 655.4769 | 698.7190 | 716.7159 | 765.8410 |
| Eryuan | 365.9509 | 434.7853 | 446.9512 | 463.0531 | 471.9941 | 590.1621 |
| Jianchuan | 312.4686 | 377.2469 | 403.1285 | 427.2047 | 436.7856 | 538.2106 |
| Heqing | 398.3198 | 406.8919 | 437.4106 | 440.9969 | 508.3256 | 571.2813 |
| Mangshi | 568.9163 | 643.5718 | 655.0890 | 658.2708 | 709.8286 | 727.3277 |
| Ruili | 585.2299 | 621.4004 | 640.0733 | 722.0235 | 729.4622 | 804.0055 |
| Lianghe | 343.3566 | 495.8395 | 512.3878 | 525.5199 | 542.1922 | 656.7071 |
| Yingjiang | 361.4221 | 362.8221 | 365.1339 | 366.4192 | 386.2395 | 472.1552 |
| Longchuan | 396.3737 | 424.4385 | 447.7375 | 450.1162 | 503.5420 | 574.8163 |
| Lushui | 338.0632 | 425.1091 | 511.3545 | 528.6956 | 621.6195 | 678.0960 |
| Fugong | 281.9702 | 301.0313 | 317.7124 | 378.3484 | 428.1135 | 499.3321 |
| Gongshan | 449.5138 | 542.8045 | 564.2108 | 653.6538 | 657.6125 | 728.4643 |
| Lanping | 379.1665 | 394.9759 | 399.0575 | 403.1246 | 427.0084 | 492.5050 |
| Shangri-La | 1421.7904 | 1484.9426 | 1498.3018 | 1568.5310 | 1588.4138 | 1616.9692 |
| Deqin | 491.5996 | 556.1296 | 606.6776 | 630.9152 | 639.8685 | 696.2319 |
| Weixi | 378.2066 | 389.1270 | 389.1924 | 403.1409 | 415.1711 | 494.6600 |

Then, we performed linear regression analysis on the comprehensive development index calculated by the traditional method and the development index calculated by the method in this paper to obtain the regression analysis result, as shown in Equation (9) and Figure 11.

$$y = 3294.3x + 275.43 \tag{9}$$

where $x$ is the development index calculated by the traditional method; $y$ is the development index calculated by the method in this paper; and $R^2$ is the correlation coefficient of the regression.

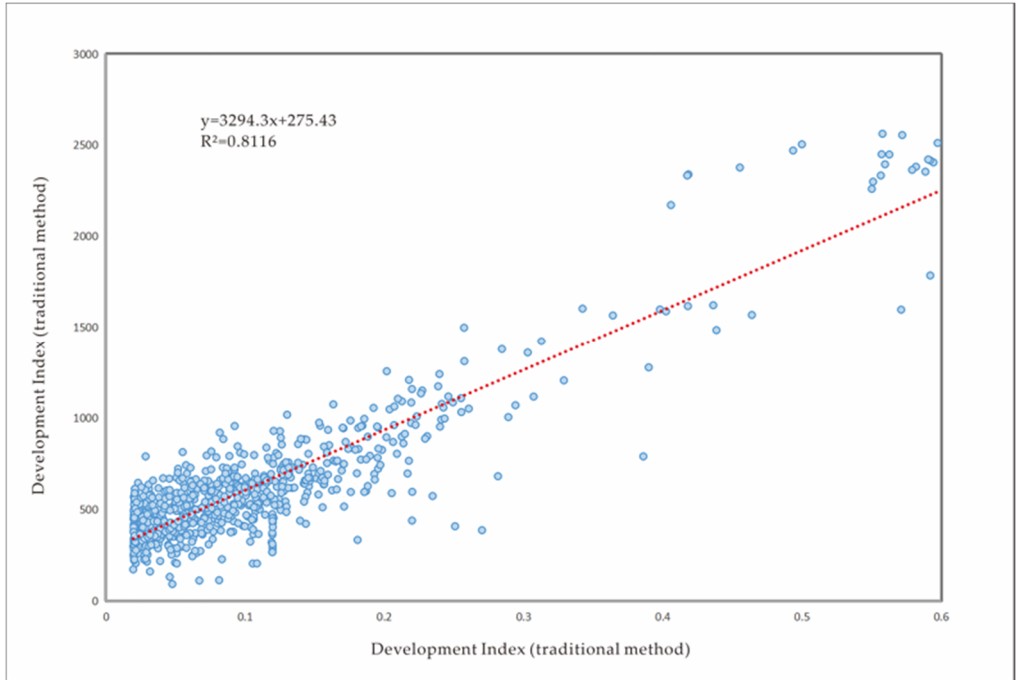

**Figure 11.** Linear regression results.

It can be seen from Figure 11 that the regression coefficient $R^2$ of the development index calculated using the method of this article and the development index calculated using the traditional method was 0.8116. When $R^2$ is greater than 0.8, it can be considered that the two variables are highly correlated. Therefore, the correctness of the method in this paper was proven.

## 4. Discussion

### 4.1. Significance to the Development of Ethnic Minorities

There are obvious differences in the development of different ethnic minorities and the development of the same ethnic minorities in different regions. This paper used the relationship between night-time light remote sensing data, economy, and population to establish the development index of ethnic minorities. The results can be analyzed by (1) the size of development differences among different ethnic minorities; (2) differences of the same minority in different minority prefectures and counties; and (3) the relationship between the development index of various ethnic minorities and geographical location. The factors in the ethnic development index model constructed in this paper can be changed, and more factors can be added according to different research purposes. This lays the foundation for the future development direction of ethnic minorities and the formulation of development policies.

Compared with the traditional research on statistical yearbooks, the method in this paper was faster, saved time, and could obtain the long-term national development status in time. In this way, we can quickly understand the development of each nation in time and space. For a multi-ethnic country, timely access to the development status of each ethnic group is conducive to adjusting policies on ethnic population, economic, and other fields to achieve coordinated and balanced development of all ethnic groups to the greatest extent, thereby reducing ethnic conflicts. The method studied in this article can not only target different ethnic groups, but can also be extended to different races and special groups (for example, using the method of this article to study the development of Blacks and Whites, and make a spatial distribution map), or different species. This is of great significance for the sustainable development and coordinated development of the world.

We used the method described in this article to calculate the development index of all ethnic groups in Yunnan Province, and used the natural discontinuity method to divide the development index into five categories. The first category was excellent-developed areas, the second category was well-developed areas, the third category was medium-developed areas, the fourth category was poor-developed areas, and the fifth category was very poor-developed areas. The classification results are shown in Figure 12.

### 4.2. The Relationship between National Development and Government

It can be roughly seen from the figure that the areas with higher national development index were mainly concentrated in the center of the county. We then counted the average distance from each type of grid to the nearest government by county. The average distance from each type of grid to the nearest government is shown in Table 6.

**Table 6.** The relationship between the level of national development and its average distance to the nearest government.

| National Development Level | Average Distance to the Nearest Government (Unit: m) |
|---|---|
| Excellent-Developed | 3352.28 |
| Well-Developed | 4695.77 |
| Medium-Developed | 6043.98 |
| Poor-Developed | 8728.84 |
| Very Poor-Developed | 12,411.90 |

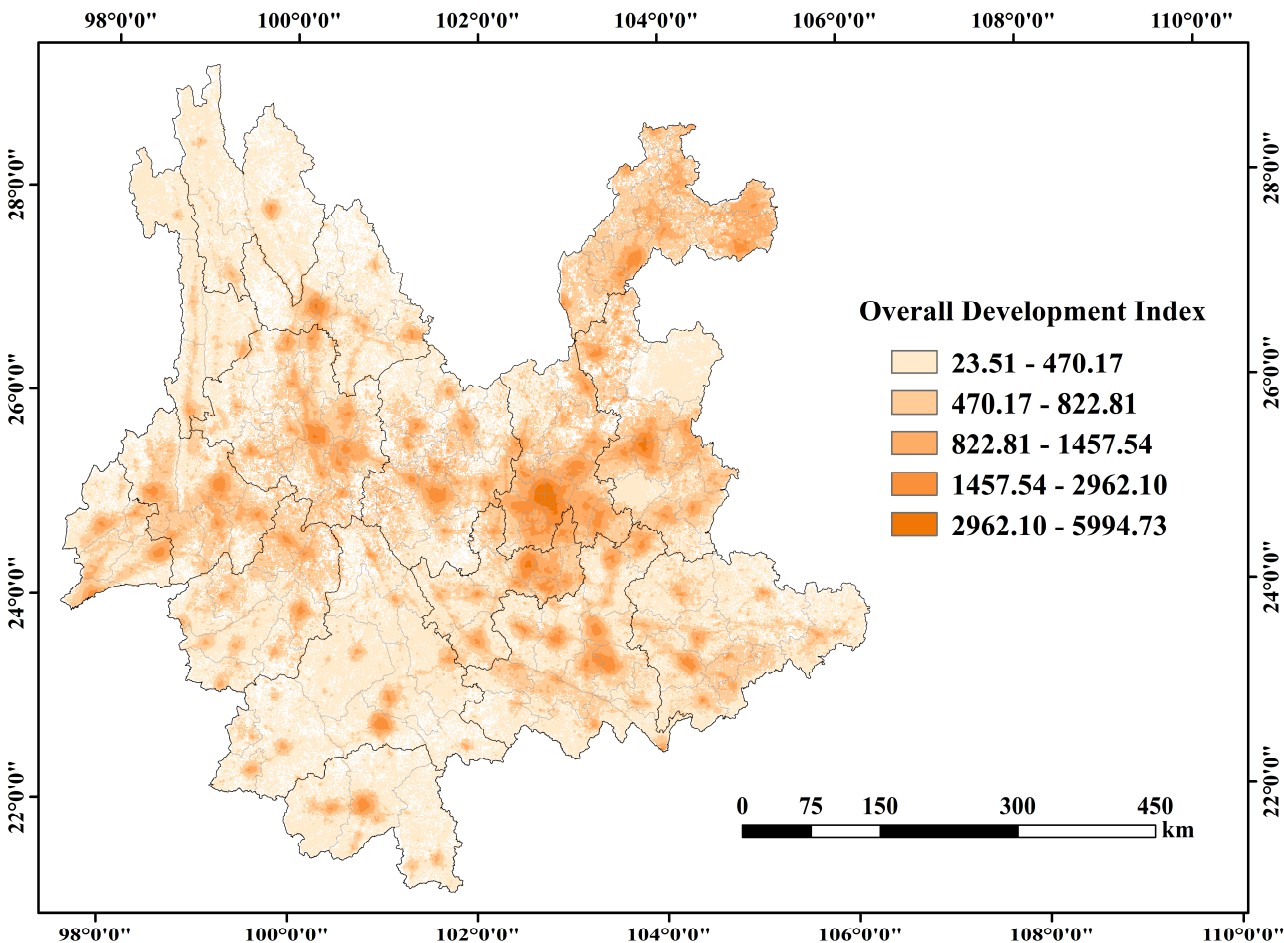

**Figure 12.** Yunnan Province nationality overall development index.

It can be seen from Table 6 that the area with excellent ethnic development is the closest to the local government. The farther the ethnic development zone is from the local government, the smaller the development index. The area with excellent ethnic development is about 3 km away from the local government, because in China, the development circle of a region is basically centered on the government and spreads around that. With the government as the center and a radius of 3 km, the higher the level of national development. With the continuous increase in the radius, the lower the level of development. Therefore, the government's assistance has played a very important role in the development of the nation.

First, we carried out regional statistics on the development index of each county, selected the average development index of each county as the benchmark, and classified the overall development index according to the county level. A grid map of the development of each county was obtained. Then, we extracted the best-developed grid center in each county, and calculated the distance between the grid center and the nearest local government. Finally, the development of each county and the distance between the best-developed areas of each county and the local government are shown on a map in Figure 13.

Generally speaking, the better-developed areas were closer to the local government. However, there were two situations on the map. First, the development of the region is better, but far from the government. The reason for this phenomenon is that the development strength of these regions is relatively strong, and the role of the government is not the main one relative to the development of the region. Second, the development of the region is poor, but is closer to the government. This phenomenon occurs because the government has not maximized its leading role in the development process of the region. In future development, we should pay attention to government assistance.

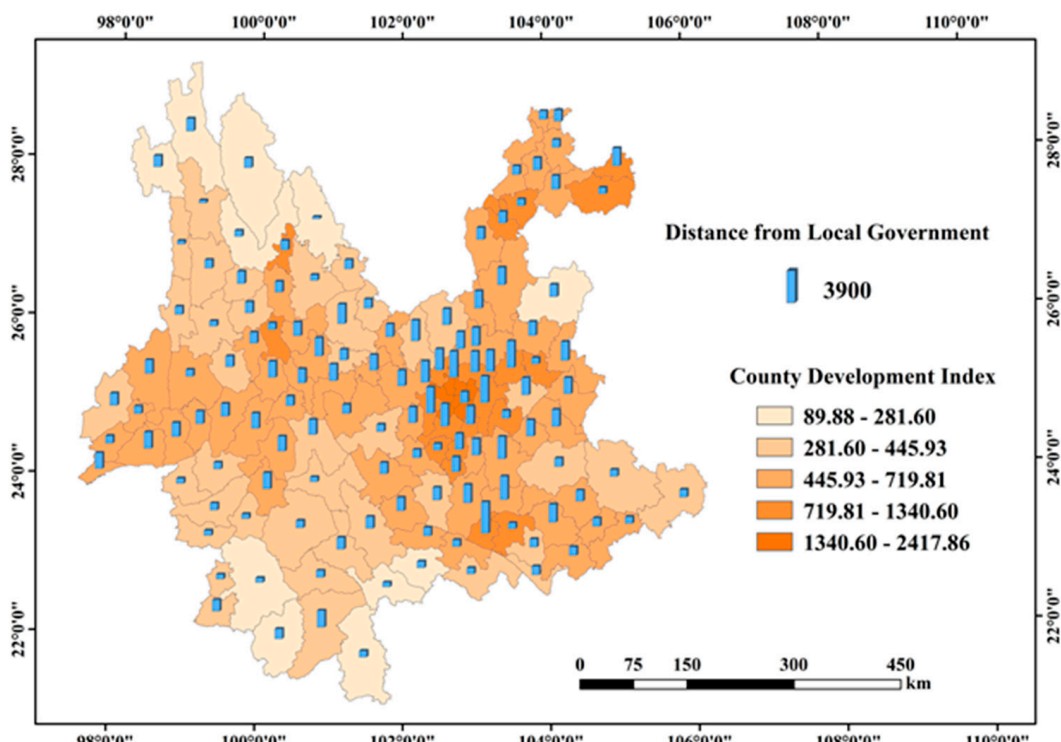

**Figure 13.** The development status of each county and the distance from the area with the highest development index of each county to the local government.

In the future development of nationalities, we must pay attention to giving play to the leading role of the government, mobilize the strength of all nationalities, and unite and assist each other in order to achieve better development.

### 4.3. Influence on Night-Time Light Remote Sensing Mapping

Night-time light remote sensing images have been widely used in economic monitoring, population mobility, environmental protection, and other fields, but there are relatively few studies [52] on night-time light remote sensing and the development of ethnic minorities. There is basically no literature on the study of ethnic minorities combined with night-time light data. This article fills this research gap to a certain extent. This paper combines toponym data, population data, and night-time light remote sensing data, considering the development of ethnic minorities from multiple perspectives. This mapping method provides a reference for subsequent similar studies. Special thematic mapping for ethnic minorities is also not common. The establishment of the ethnic minority development index plays a supporting role in dynamically monitoring the development of ethnic minorities and narrowing the development differences between ethnic minorities in various regions.

However, there are many development indexes that affect a region such as topography, population, and production patterns. Using the method in this article cannot reflect the importance of multiple variables, but can only be reflected by the brightness of night light illumination of night-time light data. The method in this article is more efficient for calculating the overall development index of a nation, but is not suitable to reflect the importance of each variable.

### 4.4. Significance of Cultural Protection of Ethnic Minorities

Due to industrialization and continuous economic development, people's production and lifestyles have undergone great changes, which has also caused many ethnic minority cultures to face crises. Therefore, we need to find the point of convergence between ethnic minority culture and economic development [55]. This article can understand the

development of ethnic minorities by establishing the minority development index, which is conducive to summarizing the development laws of ethnic minorities, and has a positive effect on the protection and inheritance of ethnic minority cultures. It also responds to the call of General Secretary Xi Jinping to pay attention to the protection and inheritance of ethnic minority cultural heritage.

## 5. Conclusions

This article used ethnic toponym data, population data, and NPP-VIIRS night-time light data to obtain the development index of each ethnic group, and analyzed the six ethnic minorities with high development index as examples. The results showed that among the six ethnic minorities, the Yi nationality had the highest development index (28.86), and the Dai had the lowest development index (15.22). After in-depth analysis, we found the relationship between the minority development index and the minority prefecture, county, and geographic location, that is, the minority development index decreased as the distance between the minority nationality and its core development prefecture and county increased. According to the obtained development indexes of ethnic minorities, combined with the toponym data of ethnic minorities, the 25 ethnic minorities were divided into 13 categories according to the language branch classification method. Each ethnic minority was classified according to the level of the development index, and a map of the distribution of ethnic minorities in Yunnan Province was obtained. The Yi were distributed in almost the entire study area, and the distribution of other ethnic minorities had obvious regional characteristics. The overall development index of ethnic minorities in the Yi, Zhuang. and Dai, and Tibetan branch was higher, and the overall development index of ethnic minorities in other language branches was lower. In most areas, multiple ethnic minorities lived together. Among them, this phenomenon was most obvious in Dehong Prefecture, which may be related to the geographical location and cultural precipitation of Dehong Prefecture. In Yunnan Province, the two ethnic minorities, Yi and Dai, live together more often with other ethnic minorities.

All in all, this paper constructed a method to calculate the development index of ethnic minorities based on NPP-VIIRS night-time light data. This method is faster and more intuitive than other qualitative analysis methods that have focused on research and statistical yearbooks. On one hand, this method makes up for the lack of corresponding economic data in rural areas and ethnic minority areas to a certain extent. On the other hand, this article provides a new idea to study the mapping of ethnic minorities and night-time light remote sensing data. This is of great significance to the development of ethnic minorities and the protection of ethnic minority culture.

**Author Contributions:** Conceptualization, F.Z.; Methodology, F.Z.; Validation, F.Z., L.S. and Z.P.; Formal analysis, L.S., Z.P., J.Y. and G.L.; Resources, Z.X.; Data curation, F.Z., L.S., J.Y. and Z.P.; Writing—original draft preparation, F.Z. and L.S.; Writing—review and editing, F.Z., Z.X., L.S., Z.P., J.Y., J.D., G.L., C.C., S.F. and Y.J.; Visualization, L.S.; Supervision, F.Z. All authors have read and agreed to the published version of the manuscript.

**Funding:** This research was funded by the National Natural Science Foundation of China (Grant No. 41961064); the Yunnan Department of Science and Technology application of basic research project (Grant No. 202001BB050030); Plateau Mountain Ecology and Earth's Environment Discipline Construction Project [Grant No.C1762101030017]; Joint Foundation Project between Yunnan Science and Technology Department and Yunnan University [Grants C176240210019]; Yunnan University Graduate Research and Innovation Fund Project (Grant No. 2020188).

**Acknowledgments:** The authors express their sincere gratitude to the Earth Observation Team, BigeMap Downloader, the National Geomatics Center of China, and the China Social Big Data Research Platform for providing the NPP-VIIRS images, POI data, toponym data, research area vector data, and census data.

**Conflicts of Interest:** The authors declare no conflict of interest.

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
