# Peer review of "Night-Time Light Remote Sensing Mapping: Construction and Analysis of Ethnic Minority Development Index"

_remotesensing, doi:10.3390/rs13112129_

Round 1

Reviewer 1 Report

In this paper, the author tried to map the ethnic minority development, which is a meaningful topic. However, the method proposed in this work is not convincing, and the analysis of the result does not fully reveal the topic. Hence, I do think the paper is not acceptable for Remote Sensing.

The main concerns are in the method section:

  • The core of the proposed model is using CPS (see equation 3) to reveal the development skill of minorities. However, the author didn’t mention the principle of how to construct CPS and why the CPS can be used to map the development of minorities. This is the most serious problem of this article.
  • One of the important parts of CPS is using kernel density analysis, but the authors don’t explain the reason to choose it. I do believe the authors have some reasons, but they need to provide them and try to convince readers. Besides, in line 212, the authors failed to explain how do they measure the effects of different search radius of KDA. Only by visual checking? 
  • Another important issue is that the author failed to separate the method and result. Please see lines 229- 239. At method section, some experiment design depends on the former experiment results. As a scientific work, the experiment design must be strictly separate from the experiment result to ensure the scientific and independent nature of the methodology.

Some other minor problems are as blows:

  • Figure 1: The map of China has different grey colors from north to south, which I believe is an error.
  • Line 121-124: the paper lacks some basic information and citations of NPP-VIIRS data. 
  • Line 167-173: the line numbers have some problems.
  • Line 179: The authors didn’t mention the resample methods of NPP-VIIRS data.
  • Figure 10: As one of the most important result figures, it is unreadable. Especially, there are 12 kinds of different labels that can hardly be distinguished. 

Author Response

Responses to Reviewer 1s Comments:

Comments and Suggestions for Authors

In this paper, the author tried to map the ethnic minority development, which is a meaningful topic. However, the method proposed in this work is not convincing, and the analysis of the result does not fully reveal the topic. Hence, I do think the paper is not acceptable for Remote Sensing.

Response: Thank you very much for your comment. I am sorry that the accuracy of the method in this article was not verified in the previous manuscript. In the revised version of the manuscript, we have added chapter 3.3 on the accuracy of the results. The development index calculated using the traditional method and the development index calculated using the method of this article are linearly regressed, and the correlation coefficient R2 is found to be 0.8116. There is a high degree of linear correlation between the two results. Therefore, the feasibility of this method is proved. At the same time, we have also enriched the writing of the introduction and discussion part.

The main concerns are in the method section:

Comment 1: The core of the proposed model is using CPS (see equation 3) to reveal the development skill of minorities. However, the author didn’t mention the principle of how to construct CPS and why the CPS can be used to map the development of minorities. This is the most serious problem of this article.

Response 1: Thank you very much for your comment. I am sorry that there is no clear description of the construction of CPS in the previous manuscript. In addition to the literature mentioned in the article, the construction of CPS in this paper also refers to the calculation form of using regular grids for sampling and calculating the population in each grid. The calculation formula of population is:

Pj=∑iAij* Dij

Where Pi is the number of people in the j grid, Aij is the total area of the ground floor of residential houses in the i census area that falls in the j grid, and Dij is the total land area of the ground floor of the residential house in the j grid. The population density of the bottom land of residential houses in the i population census area.

The formula of this article is shown in Formula 3.

The KDEi*NPPi in the formula is equivalent to the gathering area of various ethnic groups. Since the estimated value of the kernel density is too large, it is not conducive to comparison, so the formula is rooted to make the value smaller. 

Comment 2: One of the important parts of CPS is using kernel density analysis, but the authors don’t explain the reason to choose it. I do believe the authors have some reasons, but they need to provide them and try to convince readers. Besides, in line 212, the authors failed to explain how do they measure the effects of different search radius of KDA. Only by visual checking?

Response 2: Thank you very much for your comment. I’m sorry for not making it clear in the previous manuscript. The distribution of ethnic minorities in China mainly showing the distribution of “large mixed residences and small settlements”. Therefore, the toponym of ethnic minorities will be unevenly distributed, and the toponym data obtained are discrete measured values. Kernel density estimation is used to calculate the unit density of the measured value of point and line elements within a specified area. It can intuitively reflect the distribution of discrete measured values in a continuous area. Kernel density estimation can obtain the weighted average density of all data points in the study area. Therefore, we have carried out kernel density estimation on toponym data.

In addition to the results of multiple tests and visual cheaking, the choice of radius also takes into account the area of ethnic minority settlements. The area of an ethnic minority gathering area in the study area is about one square kilometer. Based on this feature, this paper sets the radius of the kernel density estimation to 1000m. It is found that the results obtained can distinguish the concentrated areas of ethnic minorities.

In the revised version of this article, we have added a corresponding statement.

Comment 3: Another important issue is that the author failed to separate the method and result. Please see lines 229- 239. At method section, some experiment design depends on the former experiment results. As a scientific work, the experiment design must be strictly separate from the experiment result to ensure the scientific and independent nature of the methodology.

Response 3: Thank you very much for your comment. I am sorry that in the previous manuscript we did not strictly distinguish between method and result. In the revised version of the manuscript, we have distinguished between the method and the result.

Some other minor problems are as blows:

Comment 1: Figure 1: The map of China has different grey colors from north to south, which I believe is an error.

Response 1: Thank you very much for your comment. I am sorry for making such a mistake. Due to our negligence, an error occurred during the export of the map of China. In the revised version of the paper, we have corrected this error.

Comment 2: Line 121-124: the paper lacks some basic information and citations of NPP-VIIRS data.

Response 2: Thank you very much for your comment. I’m sorry for not making it clear in the previous manuscript. In the revised version of the paper, we added a description of the basic information and references to the NPP-VIIRS data.

Comment 3: Line 167-173: the line numbers have some problems.

Response 3: Thank you very much for your comment. I’m sorry for this problem in the previous manuscript. In the revised version of the paper, we have corrected this problem.

 Comment 4: Line 179: The authors didn’t mention the resample methods of NPP-VIIRS data.

 Response 4: Thank you very much for your comment. I am sorry that it was not stated clearly in the previous manuscript. This article uses cubic convolution interpolation to resample the NPP-VIIRS data. In the revised version of the paper, we explained the method of NPP-VIIRS data resampling.

 Comment 5: Figure 10: As one of the most important result figures, it is unreadable. Especially, there are 12 kinds of different labels that can hardly be distinguished.

Response 5: Thank you very much for your comment. I apologize for the poor readability of the figure in the previous manuscript. In the revised version of the paper, we improved this figure.

Reviewer 2 Report

The paper entitled: “Night‐Time Light Remote Sensing Mapping: Construction and Analysis of Ethnic Minority Development Index” provides some information to justify publication. 
Specifically, it provides an interesting approach to combine different data about minority and night-time light. 
However, there is a major problem: it is not clear what the specific research questions are. The research problem and questions are not formally stated. The introduction section has not shown a literature review. It is not enough to write “got good results” (line 90). Please add conclusion or methods connected with the referenced research, papers. 
In the method section is some misunderstanding information:
•    Line 213-215 – why for the radius 1000 m the effect was the better? The better than what?  
•    Line 232-233 – why has the natural breaks method used in the classification? Why have different classifications been used for each minority? 
•    Line 248-256 – It could be better to show minority, languages in table or bullet points.

It is very good to see a lot of figures. Although, they have some mistakes:
• Figure 1. – in description (line 110) “There are 16 prefecture‐level administrative regions in Yunnan Province”. And on figure 1. the authors show the city border and county border. Please avoid blue colour to the border. Usually, the blue colour is used for rivers, water. It is better to write an abbreviation of kilometres (km).
• Figure 2. – I do not understand why Toponym data are in two different places.
• Figure 3. – What is the unit of radiance values? The high precision of radiance value is not necessary.
• Figure 4.-9. –The so high precision of the development index is not necessary.
• Figure 10. – The figure is illegible and incomprehensible. Language classification is not on the map. 
References – it is a lack of DOI.
Taking everything into account, the authors have tackled a valuable topic. If the authors organize the paper, a unique article may be created.

Author Response

Responses to Reviewer 2s Comments:

Comments and Suggestions for Authors

The paper entitled: “Night‐Time Light Remote Sensing Mapping: Construction and Analysis of Ethnic Minority Development Index” provides some information to justify publication.
Specifically, it provides an interesting approach to combine different data about minority and night-time light.
However, there is a major problem: it is not clear what the specific research questions are. The research problem and questions are not formally stated. The introduction section has not shown a literature review. It is not enough to write “got good results” (line 90). Please add conclusion or methods connected with the referenced research, papers.

Response: Thank you very much for your comment. I'm sorry that there are many imperfect descriptions in the previous manuscripts. This paper studies the development index of ethnic minorities, which can reflect the development status of each ethnic minority and the spatial distribution of the development status between ethnic groups. This plays an important role in the coordinated and balanced development of nations. In the revised version of the paper, we have enriched the writing of the introduction and discussion section, and cited more literatures. The figures in this article have been improved, and the result verification part has been added.

In the method section is some misunderstanding information:

Comment 1: Line 213-215 – why for the radius 1000 m the effect was the better? The better than what?

Response 1: Thank you very much for your comment. I am very sorry for not expressing it clearly in the previous manuscript. Because the area of an ethnic minority gathering area in the study area is about one square kilometer. According to this feature, through comparative analysis, the search radius of kernel density estimation is constantly changed, and finally it is found that when the search radius is 1000m, the effect is better, and it can distinguish ethnic minority gathering areas. And It can well reflect the gathering situation of ethnic minorities. In the revised version of this article, we have added a corresponding statement.

Comment 2: Line 232-233 – why has the natural breaks method used in the classification? Why have different classifications been used for each minority?

Response 2: Thank you very much for your comment. I am very sorry for not expressing it clearly in the previous manuscript. The natural breaks method can most appropriately group similar values and maximize the difference between each class. With reference to the paper [49], we compared the three methods of using natural breaks method, average classification method, and manual breaks method, and found that the method using natural breaks method works best.

The six ethnic minorities shown in this article all use the natural breaks method. Because the natural breaks method can maximize the difference between each category, the six ethnic minorities all use the natural breaks method to find the regional differences in the development index between each ethnic group. In this way, we can understand the development of each minority in different regions. In the revised version of the manuscript, we have added the corresponding description.

Comment 3: Line 248-256 – It could be better to show minority, languages in table or bullet points.

Response 3: Thank you very much for your comment. I apologize for not expressing the languages of ethnic minorities in a more intuitive form in the previous manuscripts. In the revised version of the manuscript, we have made corresponding adjustments.

It is very good to see a lot of figures. Although, they have some mistakes:

Comment 1: Figure 1. – in description (line 110) “There are 16 prefecture‐level administrative regions in Yunnan Province”. And on figure 1. the authors show the city border and county border. Please avoid blue colour to the border. Usually, the blue colour is used for rivers, water. It is better to write an abbreviation of kilometres (km).

Response 1: Thank you very much for your comment. I am sorry that the inappropriate color of the county boundary appeared in the previous manuscript. In the revised version of the manuscript, we have revised the color of the county boundary and used the abbreviation of kilometers (km).

Comment 2: Figure 2. – I do not understand why Toponym data are in two different places.

Response 2: Thank you very much for your comment. I'm sorry that I didn't make it clear why toponym data was used twice in the previous manuscript. The first use of toponym data is to estimate kernel density to calculate the national development index. Because some places have ethnic toponyms, but there are no ethnic minorities living in them. Taking this factor into account, the calculated ethnic development index is combined with ethnic toponyms, and those toponyms data that do not fall within the scope of the ethnic development index are eliminated to obtain the distribution range of the ethnic minority. In the revised version of the manuscript, we have added corresponding explanations.

Comment 3: Figure 3. – What is the unit of radiance values? The high precision of radiance value is not necessary.

Response 3: Thank you very much for your comment. I am very sorry for not describing it clearly in the previous manuscript. The unit of radiance values is: W/m2.μm. In the revised version of the manuscript, we only kept two decimal places.

Comment 4: Figure 4.-9. –The so high precision of the development index is not necessary.

Response 4: Thank you very much for your comment. In the revised version of the manuscript, we only kept two decimal places for the development index.

Comment 5: Figure 10. – The figure is illegible and incomprehensible. Language classification is not on the map.

Response 5: Thank you very much for your comment. I apologize for the poor readability of the figure in the previous manuscript. In the revised version of the paper, we improved this figure.

Comment 6: References – it is a lack of DOI.

Response 6: Thank you very much for your comment. I apologize for the lack of DOI in the reference paper in the previous manuscript. In the revised version of the manuscript, we have added DOI to articles that have DOI.

Reviewer 3 Report

The paper presents an interesting combination of using population and remote sensing data for the development of the Ethnic Minority Development Index. In the current state of the manuscript, it is well written. However, some details are lacking, the discussion section is really poor, as the implications of the results obtained are not discussed. In order to consider its publication, it is necessary to address the following major comments.

The following comments are listed below:

General comments

It is necessary to highlight which new contributions differ from previous studies.

The authors could improve interest for readers by highlighting what knowledge gap they are filling with their work.

What are the advantages and disadvantages of the methodology used in this study?

Are there studies similar to this one, what methodologies have been used, and how do they differ from this study?

The authors could also highlight the particular strengths and limitations of the study for possible applications of their method in other regions, contexts and scales.

For international readers, what can be learned from this study?

Include some relevant references, to enhance the discussion of the novelty of your study compared to others.

Why was Yunnan selected as a study area, e.g., because of its political role in this country, or its importance in regional ecosystems? If the ecosystem in this study area collapses, how will the disasters relate to regional, national ecosystems and climate change? What is the importance of the area?

An important issue in this quality method is to assess the robustness of the results. And here I miss a sensitive analysis: how do the results change if one variable is more important than the others?

A validation of the results is needed.

The answer to these questions should be reflected in the manuscript and should not be answered only here.

Specific comments

The introduction shows a description of the ethnic groups in China, however on a regional, continent, world scale, what is the current situation? For international readers, interested in using this methodology, it is necessary to mention it.

Line 36-38: What are these indicators? Mention and support with literature.

Line 39-42: What environments, what kind of cultures, what technologies?

Line 41-43: How does it affect directly?

Line 43-44: Could the authors explain what is unbalanced economic development? This should be reflected in the manuscript.

Line 44-45: Why does it play an important role?

Line 50: Significant differences at what level were they? <0.05, <0.0001?

Line 51-61: Not the proper way to cite, one would expect "Li [4] found....”. Check for entire manuscript. Revise instructions for author.

Line 80-82: What is that high correlation, what value was obtained?

Line 82-83: What is that good linear correlation? What value was obtained?

Line 84: Need a space in "...[16].Henderson...".

Line 84: Henderson et al. (2003) is not the correct way to cite. Revise instructions for author.

Line 87-89: What is that obvious linear relationship? What value did you get?

Line 89-90: What is that similar method? Mention

Line 91-95: What is that good linear correlation? What value did you get?

Line 95-96: What are those traditional indicators?

Line 113: Eliminate space.

Line 115: The figure needs to be improved. What are the dotted lines under China, why does China have two shades of gray?

Line 116: Remove bold font.

Line 152: Change "finally" to "Finally".

Line 156: To ensure the repeatability of this work, it is necessary to mention: How was the geometric correction performed? How was the radiometric correction performed? In what program was it performed? Is it a free or licensed software?

Line 156: In which program was the kernel density estimation performed? Is it a free or licensed software?

Line 159-160: I don't understand, the idea was left unfinished.

Line 160-161: This is not a geometric correction; it is just a transformation of the geographic coordinate system. Edit text. What is the original coordinate system of the NPP-VIIRS?

Line 164: Mention in the text.

Line 167-173: Improve presentation. Edit the arrangement of the word "Where".

Line 179-180: What is the original pixel size?

Line 182: The figure needs to be improved. A grid is needed.

Line 213: Why 1000m, how was this determined, besides multiple tests?

Line 233: The "best" according to whom? Support with literature.

Line 239: Results in materials and methods section?

Line 262-263: Same as above.

Line 268: Figure needs to be improved. A grid is needed. With 2 decimal places is enough, improve.

Line 268-279: Figure 4 to 9 should be improved. A grid is needed.

Line 284: Why is it the best? Explain.

Line 322: The figure should be improved. A grid is needed. The symbology of "Language Classification" is not shown in the territorial distribution of Yunnan province.

Line 402: What are those few studies? Cite them.

Author Response

Responses to Reviewer 3s Comments:

Comments and Suggestions for Authors

The paper presents an interesting combination of using population and remote sensing data for the development of the Ethnic Minority Development Index. In the current state of the manuscript, it is well written. However, some details are lacking, the discussion section is really poor, as the implications of the results obtained are not discussed. In order to consider its publication, it is necessary to address the following major comments.

Response: Thank you very much for your comment. I'm sorry that there are many imperfect descriptions in the previous manuscripts. In the revised version of the paper, we have enriched the writing of the introduction and discussion section, and cited more literatures. The figures in this article have been improved, and the result verification part has been added.

General comments

Comment 1: It is necessary to highlight which new contributions differ from previous studies.

Response 1: Thank you very much for your comment. I am sorry that the previous manuscript did not explain the difference between this research and previous research. Compared with the traditional research on statistical yearbooks, the research in this paper is faster, saves time, and can obtain the long-term national development status in time. In this way, we can quickly understand the development of each nation in time and space. For a multi-ethnic country, timely access to the development status of each ethnic group is conducive to adjusting policies on ethnic population, economic and other fields, so as to achieve coordinated and balanced development of all ethnic groups to the greatest extent, thereby reducing ethnic conflicts. The method studied in this article can not only target different ethnic groups, but also extend to different races and special groups, or different species. This is of great significance to the sustainable development and coordinated development of the world.

In the revised version of the manuscript, we have added a corresponding description in the discussion section of the manuscript.

Comment 2: The authors could improve interest for readers by highlighting what knowledge gap they are filling with their work.

Response 2: Thank you very much for your comment. I am very sorry for the relatively weak writing of this part in the previous manuscript. In the revised version of the manuscript, we have added a corresponding description.

Comment 3: What are the advantages and disadvantages of the methodology used in this study?

Response 3: Thank you very much for your comment. I apologize for not mentioning the advantages and disadvantages of the research in this article in the previous manuscript. The advantages of the research in this paper are: the research of development index combined with night-time light data is faster and more convenient. This paper studies the ethnic minorities and constructs a calculation method for the development index of ethnic minorities, which fills in the knowledge gap of using night-time light data to study ethnic minorities. The method in this article can not only study the development index of ethnic minorities, but also extend to different races (such as blacks and whites) and different species, which is conducive to reducing racial conflicts.

The disadvantages of this paper are: there are many development indexes that affect a region, such as topography, population, production mode, etc. However, the importance of multiple variables cannot be reflected, and can only be reflected by the luminous illumination brightness of the night-time light data. The method in this article is more efficient for calculating the overall development index of a nation, but it is not suitable to reflect the importance of each variable.

In the revised version of the manuscript, we have added corresponding statements in the discussion section.

Comment 4: Are there studies similar to this one, what methodologies have been used, and how do they differ from this study?

Response 4: Thank you very much for your comment. I am sorry that it was not stated clearly in the previous manuscript. At present, there are researches on ethnic minorities in the world, but they all use traditional research methods in statistical yearbooks. There are also studies that use night-time light data to study GDP and monitor economic activities, but they all choose a single indicator to establish its relationship with night-time light data. There is basically no literature on the study of ethnic minorities combined with night-time light data. This article fills this research gap to a certain extent. In the revised version of the manuscript, we have added a corresponding description.

Comment 5: The authors could also highlight the particular strengths and limitations of the study for possible applications of their method in other regions, contexts and scales.

Response 5: Thank you very much for your comment. I am sorry that it was not stated clearly in the previous manuscript. In the revised version of the manuscript, we have added a corresponding description in the discussion section.

Comment 6: For international readers, what can be learned from this study?

Response 6: Thank you very much for your comment. I'm sorry that in the previous manuscript, our discussion section did not describe the significance of this research for international readers. For international readers, you can use the methods of this article to study the development of different races, such as blacks and whites. This method is faster and more convenient than researching on statistical yearbook data, and can obtain the development status of ethnic groups or special groups in a long time series. In the revised version of the manuscript, we have expanded the discussion section.

Comment 7: Include some relevant references, to enhance the discussion of the novelty of your study compared to others.

Response 7: Thank you very much for your comment. I'm sorry that in the previous manuscript, our discussion section did not point out the novelty of our research with other studies. In the revised version of the manuscript, we have expanded the discussion section.

Comment 8: Why was Yunnan selected as a study area, e.g., because of its political role in this country, or its importance in regional ecosystems? If the ecosystem in this study area collapses, how will the disasters relate to regional, national ecosystems and climate change? What is the importance of the area?

Response 8: Thank you very much for your comment. I am very sorry for not describing it clearly in the previous manuscript. Among the 25 ethnic minorities in Yunnan Province, 15 ethnic minorities are unique to Yunnan, such as the Bai, Hani, Lisu, Dulong, etc. The development of ethnic minorities in Yunnan Province has made great contributions to the socio-economic development of the entire Yunnan Province. Yunnan Province is a mountainous plateau. Compared with provinces in plain areas, its topographic features are unfavorable for its development. But at the same time, Yunnan Province is located on the border of southwest China and is a key area for the development of the “Belt and Road” initiative. Therefore, choosing Yunnan as the research area of this article is representative and can also provide a certain reference for the country's economic development. In the revised version of the manuscript, we have added a corresponding description.

Comment 9: An important issue in this quality method is to assess the robustness of the results. And here I miss a sensitive analysis: how do the results change if one variable is more important than the others?

Response 9: Thank you very much for your comment. I am very sorry for not describing it clearly in the previous manuscript. If we want to consider the importance of two variables, we can take lg from the current formula:

The original formula:

The formula after taking log:

If we want to consider the degree of importance between two variables, we can determine the weight of the variable and multiply the weight value in front of the lg of each variable. In the follow-up research, we will use the deep learning method to conduct in-depth research.

Comment 10: A validation of the results is needed.

Response 10: Thank you very much for your comment. I am sorry that we did not verify the results in the previous manuscript. In the revised version of the paper, we have added a chapter 3.3 to verify accuracy.

Comment 11: The answer to these questions should be reflected in the manuscript and should not be answered only here.

Response 11: Thank you very much for your comment. For the questions in the previous manuscript, we have revised them in the revised version of the paper.

Specific comments

Comment 1: The introduction shows a description of the ethnic groups in China, however on a regional, continent, world scale, what is the current situation? For international readers, interested in using this methodology, it is necessary to mention it.

Response 1: Thank you very much for your comment. In the previous manuscripts, we did not consider the situation of ethnic minorities in the world as our shortcomings. I am very sorry. There are currently more than 2,000 ethnic groups in the world, and the total number of Asian ethnic groups is more than 1,000, accounting for about half of the total number of ethnic groups in the world. Among them, the total number of ethnic groups in China, India, the Philippines, and Indonesia exceeds 50. There are about 170 ethnic groups in Europe, and there are about 20 basically single-ethnic countries. In the revised version of the manuscript, we have added a corresponding description.

Comment 2: Line 36-38: What are these indicators? Mention and support with literature.

Response 2: Thank you very much for your comment. I am sorry that I did not clearly state the indicators for measuring the development of a region in the previous manuscript. In the revised version of the paper, we have listed these indicators and cited the corresponding references.

Comment 3: Line 39-42: What environments, what kind of cultures, what technologies?

Response 3: Thank you very much for your comment. I am very sorry for not describing it clearly in the previous manuscript. The distribution of ethnic minorities is different, and their ecological environment, cultural diversity (such as living habits, languages, religious beliefs, etc.), and production technologies (for example, some ethnic minorities still use relatively primitive production technologies) are different. These issues have been described in the revised version of the paper.

Comment 4: Line 41-43: How does it affect directly?

Response 4: Thank you very much for your comment. I am very sorry for not describing it clearly in the previous manuscript. The economic development of ethnic minorities is part of the country’s economic development and contributes to the economic development of the entire country. If there is a problem with the economic development of ethnic minorities, it will directly affect the country’s economic development to a certain extent. In the revised version of the paper, we have made a further description.

Comment 5: Line 43-44: Could the authors explain what is unbalanced economic development? This should be reflected in the manuscript.

Response 5: Thank you very much for your comment. I am very sorry for not describing it clearly in the previous manuscript. Unbalanced economic development means that due to the different ethnic groups have different cultures, settlement environments, life concepts and so on, there are different economic development models in economic development, resulting in different levels of economic development. The development of ethnic groups in some places is relatively good, and some are relatively poor, so there is a manifestation of uneven development. Unbalanced economic development has been added to the manuscript.

Comment 6: Line 44-45: Why does it play an important role?

Response 6: Thank you very much for your comment. I am very sorry for not describing it clearly in the previous manuscript. China is a multi-ethnic country, and the common development and mutual assistance of all ethnic groups can make our country stronger and more prosperous. However, due to the different levels of economic development of different ethnic groups, it is very important to understand and discover the development status of each ethnic group. This study helps to understand the development status of ethnic minorities through a simple and quick method.

Comment 7: Line 50: Significant differences at what level were they? <0.05, <0.0001?

Response 7: Thank you very much for your comment. I am very sorry for not describing it clearly in the previous manuscript. In the cited article, the urban-rural per capita income ratio of the five ethnic minority prefectures regions exceeded 2.5:1, and the highest urban-rural per capita income ratio reached 5.6:1, far exceeding the international standard (according to the general international situation, the per capita GDP is between US$800 and US$1000, and the urban-rural per capita income ratio is 1.7: 1 or so). There is a clear gap in the level of urban and rural development in ethnic minority areas. In the revised version of the manuscript, we have added relevant statements.

Comment 8: Line 51-61: Not the proper way to cite, one would expect "Li [4] found....”. Check for entire manuscript. Revise instructions for author.

Response 8: Thank you very much for your comment. I am very sorry for not using the correct way to cite the article in the previous manuscript. In the revised version of the manuscript, we corrected this problem.

Comment 9: Line 80-82: What is that high correlation, what value was obtained?

Response 9: Thank you very much for your comment. I am very sorry for not describing it clearly in the previous manuscript. When the correlation coefficient (R2) is greater than 0.8, it can be considered that the two variables are highly correlated. In the cited article, the R2 between socio-economic development and GDP reached 0.85. So it is high correlation. In the revised version of the manuscript, we have added the corresponding statement.

Comment 10: Line 82-83: What is that good linear correlation? What value was obtained?

Response 10: Thank you very much for your comment. I am very sorry for not describing it clearly in the previous manuscript. In Figure 14 of the cited article, the author studied the relationship between GDP and the area of night-time lighting in 200 countries. It can be seen from the figure that there are fewer outliers and a relatively good linear relationship.

Comment 11: Line 84: Need a space in "...[16].Henderson...".

Response 11: Thank you very much for your comment. I am sorry for the error in the previous manuscript. In the revised version of the manuscript, we have corrected this problem.

Comment 12: Line 84: Henderson et al. (2003) is not the correct way to cite. Revise instructions for author.

Response 12: Thank you very much for your comment. I apologize for such an error in the previous manuscript. In the revised version of the manuscript, we have corrected this error.

Comment 13: Line 87-89: What is that obvious linear relationship? What value did you get?

Response 13: Thank you very much for your comment. I am very sorry for not describing it clearly in the previous manuscript. In the cited article, the R2 of night light brightness and GDP is 0.8, which can be regarded as a high correlation between the two variables.

Comment 14: Line 89-90: What is that similar method? Mention

Response 14: Thank you very much for your comment. I am very sorry for not describing it clearly in the previous manuscript. The same method refers to is the same method as Henderson et al. to fit the linear relationship between GDP and night-time light data in the study area. In the revised version of the manuscript, we have improved the relevant statements.

Comment 15: Line 91-95: What is that good linear correlation? What value did you get?

Response 15: Thank you very much for your comment. I am very sorry for not describing it clearly in the previous manuscript. When the R2 is greater than 0.8, it can be considered that the two variables are highly correlated. It was found that using NPP-VIIRS night-time light data to regress with the whole city's GDP, R2 reached 0.9102. In the revised version of the manuscript, we have improved the relevant expressions.

Comment 16: Line 95-96: What are those traditional indicators?

Response 16: Thank you very much for your comment. I am very sorry for not describing it clearly in the previous manuscript. Traditional indicators refer to socio-economic parameters (such as GDP, oil and gas production, etc.). Corresponding descriptions have been added to the manuscript in the revised version.

Comment 17: Line 113: Eliminate space.

Response 17: Thank you very much for your comment. I am sorry for the mistake in the previous manuscript. In the revised version of the manuscript, we have eliminated the space.

Comment 18: Line 115: The figure needs to be improved. What are the dotted lines under China, why does China have two shades of gray?

Response 18: Thank you very much for your comment. I am very sorry for the error in the previous manuscript. The dotted line below China is China's national boundary. Due to the author's negligence, an error occurred in the process of exporting data, so two kinds of gray appeared in China. In the revised version of the manuscript, we have corrected this error.

Comment 19: Line 116: Remove bold font.

Response 19: Thank you very much for your comment. I am sorry for the mistake in the previous manuscript. I'm sorry that there was a bold font error in the previous manuscript. In the revised version of the manuscript, we have removed the bold font.

Comment 20: Line 152: Change "finally" to "Finally".

Response 20: Thank you very much for your comment. I'm sorry for the mistake of capitalization in the previous manuscript. In the revised version of the manuscript, we have corrected this error.

Comment 21: Line 156: To ensure the repeatability of this work, it is necessary to mention: How was the geometric correction performed? How was the radiometric correction performed? In what program was it performed? Is it a free or licensed software?

Response 21: Thank you very much for your comment. I am very sorry for not describing it clearly in the previous manuscript. The process of NPP-VIIRS Data Preprocessing in this study is carried out in ENVI, including geometric correction and radiometric correction. And ENVI is a licensed software. The steps of geometric correction are: load the data to be corrected in ENVI and then set the projection. This article chooses WGS_1984 geographic coordinate system. Use the geometric correction tool, set the output pixel size to 1000m, and select the cubic convolution method as the resampling method. The steps of radiometric correction are: load the data to be corrected in ENVI and use the RPC Orthorectification workflow tool for correction. First, select the average radiance value of the cloud in the low reflectivity area of the sea surface as the calibration value for removing scattered light, and then subtract the calibration value from the entire image to remove the cloud scattering. Secondly, using the method of adjacent aberrations, a threshold is set to obtain a stable surface area, and the obtained stable surface area is used as a mask, and the radiation value of the mask area is statistically analyzed. Finally, three times the average radiation value of the statistical analysis is taken as the confidence interval to remove the surface scattered light. In the revised version of the manuscript, we have added simple operations related to geometric correction and radiometric correction.

Comment 22: Line 156: In which program was the kernel density estimation performed? Is it a free or licensed software?

Response 22: Thank you very much for your comment. I am very sorry for not describing it clearly in the previous manuscript. The kernel density estimation analysis used in this study is actually performed in ArcMap, and ArcMap is a licensed software. This article uses the kernel density estimation tool in ArcMap and sets the search radius to 1000m for kernel density estimation.

Comment 23: Line 159-160: I don't understand, the idea was left unfinished.

Response 23: Thank you very much for your comment. I am sorry for the issue in the previous manuscript. Originally, what we wanted to express was: In order to avoid the influence of grid deformation, sensors and other factors on the research results, first of all, geometric correction was performed on the 2018 NPP-VIIRS monthly cloudless DNB composite data. However, when writing the manuscript, the punctuation marks were typed incorrectly, so the meaning of the sentence was not fully expressed. In the revised version of the manuscript, we revised this question.

Comment 24: Line 160-161: This is not a geometric correction; it is just a transformation of the geographic coordinate system. Edit text. What is the original coordinate system of the NPP-VIIRS?

Response 24: Thank you very much for your comment. I am very sorry for not describing it clearly in the previous manuscript. When using geometric correction tools for geometric correction in ENVI, we need to set the projection parameters. Since the geographic coordinate system of the acquired NPP-VIIRS data is WGS_1984, the projection parameter is set to WGS_1984 geographic coordinate system. In the revised version of the manuscript, we have improved the relevant expressions.

Comment 25: Line 164: Mention in the text.

Response 25: Thank you very much for your comment. I am very sorry for not describing it clearly in the previous manuscript. In the revised version of the manuscript, we have added corresponding statements.

Comment 26: Line 167-173: Improve presentation. Edit the arrangement of the word "Where".

Response 26: Thank you very much for your comment. I am sorry for the line numbers and editing errors in the previous manuscript. In the revised version of the manuscript, we have corrected this problem.

Comment 27: Line 179-180: What is the original pixel size?

Response 27: Thank you very much for your comment. The original pixel size of NPP-VIIRS data is 500m*500m. In the revised version of the manuscript, we have added relevant statements.

Comment 28: Line 182: The figure needs to be improved. A grid is needed.

Response 28: Thank you very much for your comment. In the revised version of the manuscript, we improved the picture and added a grid.

Comment 29: Line 213: Why 1000m, how was this determined, besides multiple tests?

Response 29: Thank you very much for your comment. I am very sorry for not expressing it clearly in the previous manuscript. In addition to the results of multiple tests, the choice of radius also takes into account the area of ethnic minority settlements. The area of an ethnic minority gathering area in the study area is about one square kilometer. Based on this feature, this paper sets the radius of the kernel density estimation to 1000m. It is found that the results obtained can distinguish the concentrated areas of ethnic minorities. In the revised version of this article, we have added a corresponding statement.

Comment 30: Line 233: The "best" according to whom? Support with literature.

Response 30: Thank you very much for your comment. I am very sorry for not expressing it clearly in the previous manuscript. We refer to the literature [49], they use the natural break method to get better results. So we compared the three methods using natural break, average classification, and manual break, and found that the natural break method works best. In the revised version of the manuscript, we have added relevant descriptions.

Comment 31: Line 239: Results in materials and methods section?

Response 31: Thank you very much for your comment. I am sorry that in the previous manuscript we did not strictly distinguish between method and result. In the revised version of the manuscript, we have distinguished between the method and the result.

Comment 32: Line 262-263: Same as above.

Response 32: Thank you very much for your comment. I am sorry that in the previous manuscript we did not strictly distinguish between method and result. In the revised version of the manuscript, we have distinguished between the method and the result.

Comment 33: Line 268: Figure needs to be improved. A grid is needed. With 2 decimal places is enough, improve.

Response 33: Thank you very much for your comment. In the revised version of the manuscript, we improved the picture and added a grid. And only two decimal places are kept.

Comment 34: Line 268-279: Figure 4 to 9 should be improved. A grid is needed.

Response 34: Thank you very much for your comment. In the revised version of the manuscript, we improved the picture and added a grid.

Comment 35: Line 284: Why is it the best? Explain.

Response 35: Thank you very much for your comment. I am very sorry for not expressing it clearly in the previous manuscript. Because the development index calculated in this article is a comprehensive development index. From the calculation results, it can be known that the development index of the Yi nationality is the highest, and there are many areas with high development indexes, so the development of the Yi nationality is the best. In the revised version of the manuscript, we have added a corresponding description.

Comment 36: Line 322: The figure should be improved. A grid is needed. The symbology of "Language Classification" is not shown in the territorial distribution of Yunnan province.

Response 36: Thank you very much for your comment. I apologize for the poor readability of the figure in the previous manuscript. In the revised version of the paper, we improved this graph and added a grid.

Comment 37: Line 402: What are those few studies? Cite them.

Response 37: Thank you very much for your comment. I am very sorry for not expressing it clearly in the previous manuscript. In the revised version of the paper, we cite these articles about ethnic minorities.

Round 2

Reviewer 1 Report

I am satisfied with the improvement of the manuscript.

Author Response

Thank you very much for your comment on our paper " Night-Time Light Remote Sensing Mapping: Construction and Analysis of Ethnic Minority Development Index " (ID: remotesensing-1212746). These comments help us improve the quality of this article. These comments are very valuable, and will also be of great help for revising and improving our thesis. Thanks again for your comments.

Reviewer 2 Report

I accept in present form

Author Response

Thank you very much for your comment on our paper " Night-Time Light Remote Sensing Mapping: Construction and Analysis of Ethnic Minority Development Index " (ID: remotesensing-1212746). These comments are very valuable, these comments improve the quality of this article, and have an important guiding role in revising and improving our article. Thanks again for your comments.

Reviewer 3 Report

The authors made a significant effort. No further comments.

Author Response

Thank you very much for your comment on our paper " Night-Time Light Remote Sensing Mapping: Construction and Analysis of Ethnic Minority Development Index " (ID: remotesensing-1212746). These comments have improved the quality of this article, and have an important guiding role in revising and improving our article. Thanks again for your comments.